# REVISITING HIGHER-ORDER GRADIENT METHODS FOR MULTI-AGENT REINFORCEMENT LEARNING

## ABSTRACT

This paper revisits Higher-Order Gradient (HOG) methods for Multi-Agent Reinforcement Learning (MARL). HOG methods are algorithms in which agents use higher-order gradient information to account for other agents' anticipated learning, and are shown to improve coordination in games with self-interested agents. So far, however, HOG methods are only applied to games with low-dimensional state spaces due to inefficient computation and preservation of higher-order gradient information. In this work, we solve these limitations and propose a HOG framework that can be applied to games with higher-dimensional state spaces. Moreover, we show that current HOG methods, when applied to games with common-interested agents, i.e., team games, can lead to miscoordination among the agents. To solve this, we propose Hierarchical Reasoning (HR) to improve coordination in team games, and we experimentally show that our proposed HR significantly outperforms state-of-the-art methods in standard multi-agent games. With our contributions, we greatly improve the applicability of HOG methods for MARL. For reproducibility, the code used for our work will be shared after the reviewing process.

## 1 INTRODUCTION

In multi-agent systems, the paradigm of *agents' reasoning about other agents* has been explored and researched extensively (Goodie et al., 2012; Liu & Lakemeyer, 2021). Recently, this paradigm is being studied in the subfield of Multi-Agent Reinforcement Learning (MARL) (Wen et al., 2019; 2020; Konan et al., 2022). Generally speaking, MARL deals with several agents simultaneously learning and interacting in an environment. In the context of MARL, reasoning can be interpreted as accounting for the anticipated learning of other agents (Zhang & Lesser, 2010). As MARL uses gradient-based optimization, learning anticipation naturally leads to the usage of higher-order gradient information (Letcher et al., 2019). The so-called Higher-Order Gradient (HOG) methods use this extra gradient information to predict and, in some cases, shape the learning of other agents (Letcher et al., 2019). The importance of prediction and shaping has been frequently shown for various games, such as the Iterated Prisoner's Dilemma (IPD), where shaping ensures cooperation among the agents (Foerster et al., 2018a). However, current HOG methods have clear limitations, as they can only work for specific types of games, and become inefficient when the dimensionality of the game increases. In this paper, we explore these limitations and propose a framework that can extend the application scope of HOG methods to a broader range of problem settings in MARL.

The vast majority of existing HOG methods focus only on games with low-dimensional state spaces, e.g., matrix games (Foerster et al., 2018a;b; Willi et al., 2022). There are two challenges that limit HOG methods from being applied to games with high-dimensional state spaces: inefficient computation and preservation of higher-order gradient information. Specifically, current implementations of HOG methods require multiple data sampling stages to compute higher-order gradient information (Foerster et al., 2018b). Moreover, the higher-order gradient information is applied and, more importantly, preserved in the policy network's parameter space. As a result, existing HOG methods become very inefficient when applied to games that have high-dimensional state spaces, and therefore require high-dimensional parameter spaces. In this paper, to solve this, we propose an HOG framework where the higher-order gradient information are computed and preserved more efficiently. By comparing our proposed framework to existing HOG methods in well-controlled studies, we demonstrate that the overall performance and efficiency of our proposed framework stay

consistent with increased dimensionality, unlike for existing HOG methods, where they get drastically worse.

In addition to dimensionality limitations, the generalizability of HOG methods to various types of games is questionable. Originally, HOG methods are proposed to improve cooperation in games with self-interested agents (Zhang & Lesser, 2010; Foerster et al., 2018a). So far, however, it is unclear how HOG methods perform when agents are fully cooperative, i.e., for common-interested agents in team games. We demonstrate that existing HOG methods have the tendency to lead to miscoordination among common-interested agents, causing a sub-optimal overall reward. To solve this, and improve the applicability of HOG methods to team games, we propose Hierarchical Reasoning (HR), a new HOG methodology explicitly developed for improving coordination in games with common-interested agents. Below, we summarize our contributions.

- We propose HOG-MADDPG, a framework to make existing HOG methodologies, e.g., LA and LOLA, applicable to games with higher-dimensional state spaces by solving the limitations in computation and preservation of higher-order gradient information. With our framework, we develop two novel HOG methods, LA-MADDPG and LOLA-MADDPG, which apply the principles of LA and LOLA, respectively.
- We demonstrate theoretically, in a two-agent two-action coordination game, and empirically, in a two-agent three-action coordination game, that the existing HOG methodologies can suffer from miscoordination among common-interested agents. To solve this, we propose the HR methodology and show, theoretically and empirically, that it overcomes miscoordination in the coordination games.
- We apply the HR principle to our HOG-MADDPG framework and develop HR-MADDPG, a HOG method for common-interested agents. We show that HR-MADDPG outperforms the existing state-of-the-art methods on standard multi-agent games.

## 2 RELATED WORKS

When direct communication among agents is not possible, the standard tool for MARL agents to apply reasoning is Agents Modeling Agents (AMA) (Albrecht & Stone, 2018). Although agents traditionally use AMA to only predict the behavior of others (He et al., 2016; Hong et al., 2018), recent studies have extended AMA to further consider multiple levels of reasoning over the predicted behaviors (Wen et al., 2019; 2020). However, these approaches do not explicitly account for the other agents' anticipated learning, which has shown to be beneficial in games where interaction among self-interested agents naturally leads to worst-case outcomes (Foerster et al., 2018a).

HOG methods, on the other hand, are a range of methods that use higher-order gradient information to predict and, in some cases, shape the anticipated learning of other agents directly. This includes Learning with Opponent-Learning Awareness (LOLA), proposed to shape opponents for better coordination in Iterated Prisoner's Dilemma (IPD) by Foerster et al. (2018a), Look-Ahead (LA), proposed to guarantee convergence in cyclic games by Zhang & Lesser (2010), Stable Opponent Shaping (SOS), developed by Letcher et al. (2019) as an interpolation between LOLA and LA to inherit the benefits of both, and other methods such as Consensus Optimization (CO) and Symplectic Gradient Adjustment (SGA), that are proposed to improve cooperation by Bertsekas (2014) and Balduzzi et al. (2018), respectively. However, as we explain in Section 4, these methods have only been applied to simple games due to the challenges in computation and preservation of higher-order gradient information. Furthermore, the impact of current HOG methods on coordination among common-interested agents has not yet been fully investigated. Current investigations are limited to convergence and non-convergence to stable and unstable fixed points in differential games, respectively (Letcher et al., 2019). However, we demonstrate in Section 5.1 that in the case of a two-agent, two-action coordination game with unstable fixed points, HOG methods can converge to miscoordination points. The focus of this work is to extend current HOG methodology so that it can be used for games with higher-dimensional state spaces and common-interested agents.

## 3 BACKGROUND

We formulate the MARL setup as a Markov Game (MG) (Littman, 1994). An MG is a tuple $(\mathcal{N}, \mathcal{S}, \{\mathcal{A}_i\}_{i \in \mathcal{N}}, \{\mathcal{R}_i\}_{i \in \mathcal{N}}, \mathcal{T}, \rho, \gamma)$, where $\mathcal{N}$ is the set of agents ($|\mathcal{N}| = n$), $\mathcal{S}$ is the set of states,

and $\mathcal{A}_i$ is the set of possible actions for agent $i \in \mathcal{N}$. Agent $i$ chooses its action $a_i \in \mathcal{A}_i$ through the policy network $\pi_{\theta_i} : \mathcal{S} \times \mathcal{A}_i \to [0,1]$ parameterized by $\theta_i$ conditioning on the given state $s \in \mathcal{S}$. Given the actions of all agents, each agent $i$ obtains a reward $r_i$ according to its reward function $\mathcal{R}_i : \mathcal{S} \times \mathcal{A}_1 \times ... \times \mathcal{A}_n \to \mathbb{R}$. Given an initial state, the next state is produced according to the state transition function $\mathcal{T} : \mathcal{S} \times \mathcal{A}_1 \times ... \times \mathcal{A}_n \to \mathcal{S}$. We denote an episode of horizon $T$ as $\tau = (\{s^0, a_1^0, ..., a_n^0, r_1^0, ..., r_n^0\}, ..., \{s^T, a_1^T, ..., a_n^T, r_1^T, ..., r_n^T\})$, and the discounted return for each agent $i$ at time step $t \leq T$ is defined by $G_i^t(\tau) = \sum_{l=t}^T \gamma^{l-t} r_i$ where $\gamma$ is a predefined discount factor. The expected return given the agents' policy parameters approximates the state value function for each agent $V_i(s, \theta_1, ..., \theta_n) = \mathbb{E}[G_i^t(\tau|s^t = s)]$. Each agent $i$ aims to maximize the expected return given the distribution of the initial state $\rho(s)$, denoted by the performance objective $J_i = \mathbb{E}_{\rho(s)} V_i(s, \theta_1, ..., \theta_n)$. A *naïve agent* updates its policy parameters in the direction of the objective's gradient: $\nabla_{\theta_i} J_i = \mathbb{E}_{\rho(s)} \nabla_{\theta_i} V_i(s, \theta_1, ..., \theta_n)$.

**Learning With Opponent-Learning Awareness (LOLA)**. Unlike naïve agents, LOLA agents modify their learning objectives by differentiating through the anticipated learning steps of the opponents (Foerster et al., 2018a). Given $n = 2$ for simplicity, a first-order LOLA agent assumes a naïve opponent and optimizes $V_1^{\text{LOLA}}(s, \theta_1, \theta_2 + \Delta\theta_2)$ where $\Delta\theta_2 = \eta\nabla_{\theta_2} V_2(s, \theta_1, \theta_2)$ and $\eta$ is the prediction length. Using first-order Taylor expansion, and by differentiating with respect to $\theta_1$, the gradient adjustment for the first LOLA agent (Foerster et al., 2018a) is given by

$$\nabla_{\theta_1} V_1^{\text{LOLA}}(s, \theta_1, \theta_2 + \Delta\theta_2) \approx \nabla_{\theta_1} V_1 + (\nabla_{\theta_2\theta_1} V_1)^\intercal \Delta\theta_2 + \underbrace{(\nabla_{\theta_1} \Delta\theta_2)^\intercal \nabla_{\theta_2} V_1}_{\text{shaping}}, \tag{1}$$

where $V_1 = V_1(s, \theta_1, \theta_2)$. The rightmost term in the LOLA update allows for active shaping of the opponent's learning. This term has been proven effective in enforcing cooperation in various games, including IPD (Foerster et al., 2018a;b). The LOLA update can be further extended to non-naïve opponents, resulting in higher-order LOLA agents (Foerster et al., 2018a; Willi et al., 2022).

**Look Ahead (LA).** LA agents assume that the opponents' learning steps cannot be influenced, i.e., cannot be shaped (Zhang & Lesser, 2010; Letcher et al., 2019). In other words, agent 1 assumes that the prediction step, $\Delta\theta_2$, is independent of the current optimization, i.e., $\nabla_{\theta_1} \Delta\theta_2 = 0$. Therefore, the shaping term disappears and the gradient adjustment for the first LA agent will be

$$\nabla_{\theta_1} V_1^{\text{LA}}(s, \theta_1, \theta_2 + \Delta\theta_2) \approx \nabla_{\theta_1} V_1 + (\nabla_{\theta_2\theta_1} V_1)^\intercal \Delta\theta_2. \tag{2}$$

# 4 A HOG FRAMEWORK FOR HIGH-DIMENSIONAL STATE SPACES

Existing HOG methods like LOLA and LA are only applied to games with low-dimensional state spaces, e.g., matrix games (Zhang & Lesser, 2010; Foerster et al., 2018b;a; Letcher et al., 2019; Willi et al., 2022). When applied to games with higher-dimensional state spaces, they become very inefficient, due to the way the higher-order gradient information is computed and preserved. In this section, we analyze these problems, and we propose a framework that makes HOG practical for application to games with high-dimensional state spaces.

## 4.1 LIMITATIONS OF EXISTING HOG APPROACHES

**Computation of higher-order gradient**. Existing HOG methods are implemented in the stochastic policy gradient framework, and optimize non-differentiable objectives. Furthermore, the learning step for one agent in the standard stochastic policy gradient theorem is independent of other agents' parameters. Therefore, higher-order mixed partial derivatives (among multiple agents) cannot be easily computed. Foerster et al. (2018b) proposed an infinitely differentiable Monte Carlo estimator, referred to as DiCE, to correctly optimize the stochastic objectives with any order of gradients. Similarly to meta-learning frameworks, the agents reason about and predict the learning process of the opponents using inner learning loops and update their parameters in outer learning loops. However, each learning loop for each agent requires a sampling stage which is very inefficient for high-order reasoning and games with higher-dimensional state spaces, i.e., beyond matrix games.

**Preservation of higher-order gradient information**. HOG methods should constantly compute and update the higher-order gradient values and computation graphs that keep track of how the

gradients should flow. In the implementation of existing HOG methods, the higher-order gradient information is computed and preserved in the parameter spaces of the agents' policy networks (Foerster et al., 2018a;b). As a result, the agents should either have access to other agents' exact parameters or infer other agents' parameters from state-action trajectories (Foerster et al., 2018a). In many game settings, these parameters are obscured. Moreover, when the dimensionality of state spaces increases, e.g., when having images as input, the dimensionality of the parameter spaces increases as well, making the parameter inference problem computationally expensive. Furthermore, computing and storing the higher-order gradient information in high-dimensional parameter spaces is inefficient, whether the parameters are inferred or exact.

## 4.2 HOG-MADDPG

To efficiently compute any-order mixed partial derivatives, we need to optimize differentiable objectives, which are directly dependent on all agents' parameters and can be efficiently estimated. The only platform that meets the above requirements and can deal with both discrete and continuous action spaces is Multi-Agent Deep Deterministic Policy Gradient (MADDPG) (Lowe et al., 2017). In this platform, decentralized policies are trained to optimize centralized, differentiable objectives, i.e., state-action value functions, that are estimated efficiently from trajectories sampled from a distinct behavior policy, i.e., the off-policy approach. Therefore, we propose to build our HOG framework on top of the MADDPG platform. Similarly to MADDPG, we follow the Centralized Training and Decentralized Execution (CTDE) setting in our work. However, differently from MADDPG, we better exploit the available information in CTDE by accounting for the agents' anticipated learning.

The only unsolved problem is the preservation of higher-order gradient information, as the centralized learning of MADDPG does not grant access to agents' parameters. To solve this, we propose to project the anticipated gradient information from the policies' parameter spaces to the action spaces. This way, 1) we avoid additional constraints above the centralized state-action value functions where the agents have access to all actions, and 2) we improve efficiency as the action spaces have significantly lower dimensionality than the policies' parameter spaces. In Appendix B we theoretically analyze the influence of the proposed projection concept on the overall performance (Appendix B.1), and the time complexity of gradient anticipation (Appendix B.2). In the following sections, we apply our proposed framework to two HOG methods and explain the details of their update rules for policy parameters. For all proposed methods in the HOG-MADDPG frameworks, the centralized state-action value functions are updated in a way identical to MADDPG (Lowe et al., 2017).

### 4.2.1 LA-MADDPG

In the MADDPG platform, a deterministic policy $\mu_{\theta_i}$ for agent $i$ is defined as $\mu_{\theta_i} : \mathcal{S} \to \mathcal{A}_i$, parameterized by $\theta_i$. By denoting a centralized state-action function for each agent $i$ as $Q_i(s, a_1, ..., a_n) = \mathbb{E}[G_i^t(\tau|s^t = s, a_i^t = a_i \forall i \in \mathcal{N})]$, the gradient of the MADDPG performance objective $J_i$ with respect to $\theta_i$ can be approximated as:

$$\nabla_{\theta_i} J_i \approx \mathbb{E}_{\rho^\beta(s,\hat{a})} \nabla_{\theta_i} Q_i(s, \hat{a}_1, ..., a_i, ..., \hat{a}_n)|_{a_i = \mu_{\theta_i}(s)}, \tag{3}$$

where $\rho^\beta(s, \hat{a})$ is the state-action distribution of the behavior policy and $\hat{a} = \{\hat{a}_i \forall i \in \mathcal{N}\}$ are the actions sampled from the behavior policy during the exploration stage. Given $n = 2$ and a naïve opponent for simplicity, the gradient adjustment for parameters of the LA-MADDPG agent ($\theta_1$) is computed by accounting for the anticipated policy parameters of the opponent, i.e., $\hat{\theta}_2 + \Delta\hat{\theta}_2(s)$:

$$\nabla_{\theta_1} J_1^{\text{LA}} \approx \mathbb{E}_{\rho^\beta(s)} \nabla_{\theta_1} Q_1(s, a_1, \tilde{a}_2)|_{a_1 = \mu_{\theta_1}(s), \tilde{a}_2 = \mu_{\hat{\theta}_2 + \Delta\hat{\theta}_2}(s)}, \tag{4}$$

where $\Delta\hat{\theta}_2 = \eta \nabla_{\hat{\theta}_2} Q_2(s, \hat{a}_1, \hat{a}_2)|_{\hat{a}_1 = \mu_{\hat{\theta}_1}(s), \hat{a}_2 = \mu_{\hat{\theta}_2}(s)}$, and $\hat{\theta}_1$ and $\hat{\theta}_2$ are the behavior policy parameters. As the agents cannot have access to these parameters, we propose to project the anticipated gradients to the action space (see Appendix A.1):

$$\nabla_{\theta_1} J_1^{\text{LA}} \approx \mathbb{E}_{\rho^\beta(s,\hat{a})} \nabla_{\theta_1} \mu_{\theta_1}(s) \nabla_{a_1} Q_1(s, a_1, \hat{a}_2 + \Delta\hat{a}_2)|_{a_1 = \mu_{\theta_1}(s)}, \tag{5}$$

where $\Delta\hat{a}_2 = \hat{\eta} \nabla_{\hat{a}_2} Q(s, \hat{a}_1, \hat{a}_2)$ and $\hat{\eta}$ is the projected prediction length (see Alg. 1).

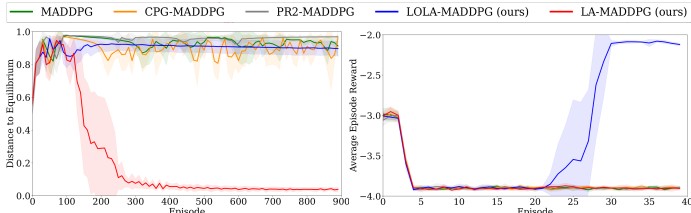

| Method | IRG (DtE ↓) |
|---|---|
| LA-DiCE | 0.09±0.07 |
| LA-MADDPG (ours) | **0.03±0.02** |

| Method | IPD (AER ↑) |
|---|---|
| LOLA-DiCE | -2.16±0.12 |
| LOLA-MADDPG (ours) | **-2.08±0.02** |

Figure 1: Learning curves in two matrix games. Left: Iterated Rotational Game in terms of the distance to the equilibrium point (↓). Right: Iterated Prisoner's Dilemma in terms of the normalized averages return (↑).

Table 1: Comparisons of the LA and LOLA methods in the frameworks of DiCE and our proposed HOG-MADDPG.

### 4.2.2 LOLA-MADDPG

In the standard MADDPG, it is assumed that the opponents' actions are fixed during the optimization steps for the agents. This assumption is problematic for LOLA agents that require shaping on the opponents' learning steps. To make these dependencies possible, we employ the Centralized Policy Gradient (CPG) (Peng et al., 2021) to derive the learning step of the LOLA agents. In CPG-MADDPG, the gradient update in Eq. (3) is modified to:

$$\nabla_{\theta_i} J_i \approx \mathbb{E}_{\rho^\beta(s)} \nabla_{\theta_i} Q_i(s, a_1, ..., a_n)|_{a_i = \mu_{\theta_i}(s)} \forall i \in \mathcal{N}. \tag{6}$$

Again, we propose to project the anticipated gradient information to the action space. Given $n = 2$, the first-order LOLA-MADDPG agent updates the policy parameters through (see Appendix A.2):

$$\nabla_{\theta_1} J_1^{\text{LOLA}} \approx \mathbb{E}_{\rho^\beta(s)} \nabla_{\theta_1} \mu_{\theta_1}(s) \nabla_{a_1} Q_1(s, a_1, a_2 + \Delta a_2)|_{a_1 = \mu_{\theta_1}(s), a_1 = \mu_{\theta_2}(s)}, \tag{7}$$

where $\Delta a_2 = \hat{\eta} \nabla_{a_2} Q(s, a_1, a_2)$, which, unlike for LA-MADDPG, is a function of $a_1$ (see Alg. 2).

### 4.3 EXPERIMENTS

In this section, we first verify our proposed HOG-MADDPG methods on simple matrix games, to show how they work in the same situations as the original HOG methods. Second, we apply our proposed methods to games with high-dimensional state spaces, which is the envisioned use case for our methods, and evaluate the performance and efficiency. As the primary baselines, we apply HOG methods on the DiCE framework (Foerster et al., 2018b), i.e., LA-DiCE and LOLA-DiCE. We further compare our proposed methods with the standard MADDPG configured with three state-of-the-art update rules: 1) standard update rule (Lowe et al., 2017), referred to as MADDPG, 2) CPG update rule (Peng et al., 2021), referred to as CPG-MADDPG, and 3) Probabilistic Recursive Reasoning (PR2) update rule (Wen et al., 2019), referred to as PR2-MADDPG. For a fair comparison, we have employed identical architectures with the same number of policy and value function parameters for all the baseline (see Appendix D).

### 4.3.1 MATRIX GAMES

We evaluate the methods on the following commonly-used, two-agent matrix games: 1) Iterated Rotational Game (IRG) (Zhang & Lesser, 2010), a one-state game with a 1-Dimensional (1-D) continuous action space between 0 and 1 representing the probability of taking two discrete actions, and 2) Iterated Prisoner's Dilemma (IPD) (Foerster et al., 2018a), a five-state game with two discrete actions and $T = 150$. The games are developed to highlight the strengths of specific HOG methods (IRG for LA-based methods and IPD for LOLA). Further details about these games are provided in Appendix D.1. We evaluate the performances of methods based on the Distance to Equilibrium (DtE) in IRG (the equilibrium point in IRG is reached when $a_1 = a_2 = 0.5$) and the Averaged Episode Reward (AER) in IPD. In Figure 1, we depict the learning curves for our methods and other, state-of-the-art MADDPG-based algorithms. From this figure, we find that our HOG methods are the only MADDPG-based networks that can effectively solve these games, with LA-MADDPG for IRG and LOLA-MADDPG for IPD. This highlights the importance of using higher-order gradient information. To further show its effectiveness, we compare with current DiCE-based HOG methods (Foerster et al., 2018b) which are designed for these matrix games, in Table 1, and we find that we even achieve slightly better results.

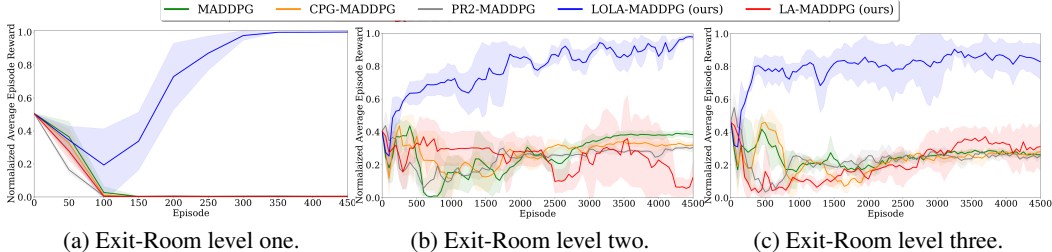

(a) Exit-Room level one.      (b) Exit-Room level two.      (c) Exit-Room level three.

Figure 2: Learning curves in different complexity levels of the exit-room game in terms of the normalized average return. Higher values are better.

| Methods | ↑NAER in Exit-Room game | | | ↓NTTI in Exit-Room game | | | | |
|---|---|---|---|---|---|---|---|---|
| | $l=1$ | $l=2$ | $l=3$ | Naïve | 1st-order | 2nd-order | 3rd-order | 4th-order |
| LOLA-DiCE | 0.91±0.04 | 0.68±0.06 | 0.56±0.12 | 1 | 2.39 | 3.74 | 5.12 | 6.41 |
| LOLA-MADDPG (ours) | **1.00±0.00** | **0.99±0.01** | **0.93±0.03** | 1 | **1.03** | **1.05** | **1.08** | **1.12** |

Table 2: Comparisons of DiCE with our proposed HOG-MADDPG in the Exit-Room game, in terms of performance (normalized average return in different game levels) and efficiency (training time per iteration in different reasoning levels).

### 4.3.2 MULTI-LEVEL EXIT-ROOM GAME

Inspired by Vinitsky et al. (2019), we propose an Exit-Room game with three levels of complexity (see Figure 3). The Exit-Room game is a grid-world variant of the IPD, with two agents (blue and red), and is specifically developed to highlight the strength of LOLA. The agents should cooperate and move towards the exit doors on the right. However, they are tempted to exit the left doors, and in some cases, not exiting at all. In level 1, the agents have three possible actions (*move-left*, *move-right*, or *do nothing*), and in levels 2 and 3, they have additional *move-up* and *move-down* actions. Additionally, in level 3, the door positions are randomly located, resulting in more complex interactions among the agents. For more details about the game, see Appendix D.2.

Figure 2 compares the learning curves of LOLA-MADDPG with the state-of-the-art, MADDPG-based methods in terms of Normalized Average Episode Reward (NAER) which is the AER value that is normalized between the highest and lowest episode rewards in each game level. In Figure 2, we can clearly see that our LOLA-MADDPG significantly outperforms the other methods, similarly as for the matrix games. To highlight the benefits of our proposed method with respect to existing HOG methods, we compare our LOLA-MADDPG

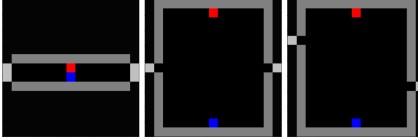

Figure 3: State observation in the Exit-Room game, level one (left), level two (middle), and level three (right).

with LOLA-DiCE in terms of performance (by comparing NAER) and training efficiency, in Table 2. For training efficiency, we use the average Training Time per Iteration (TTI) for various opponents' reasoning levels. For a fair comparison, we report Normalized TTI (NTTI) for both methods, which are the normalized TTI values with respect to naïve (zero-order) version of each method. Observing Table 2, it is apparent that LOLA-DiCE fails to acquire the highest rewards, particularly for the second and third levels of the game, where complexity is increased. Moreover, our proposed LOLA-MADDPG performs better in all levels of the game in terms of NAER, and scaling from naïve to higher-order opponents is significantly more efficient for LOLA-MADDPG than LOLA-DiCE. This emphasizes that we have overcome the limitations of HOG methods described in Section 4.1.

## 5 A HOG METHODOLOGY FOR COMMON-INTERESTED AGENTS

Current HOG methods are proposed to enforce coordination in games with self-interested agents. In many applications, however, agents should cooperate to increase a common reward function, i.e., as a team game. As multiple agents should interact and cooperate, anticipating the learning of other agents, which is the core idea of HOG methods, has the potential to work well in these types of games too. In this section, we first show that standard HOG methods do not work well for team games, because they suffer from miscoordination. Subsequently, we propose a method to overcome this limitation, and evaluate it for several different games.

### 5.1 MISCOORDINATION ANALYSIS IN COOPERATIVE SCENARIOS

To investigate the effectiveness of HOG methods on the coordination among common-interested agents, we consider a two-agent, two-action coordination game (Claus & Boutilier, 1998) with an added miscoordination penalty. The game is defined by a common reward matrix $R = \begin{bmatrix} a & k \\ k & a \end{bmatrix}$, where $a > 0$ is coordination reward and $k \leq 0$ is the miscoordination penalty. We further define $g = a - k > 0$ as the miscoordination regret. The agents are parameterized by $\theta_1 \in [0, 1]$ and $\theta_2 \in [0, 1]$, denoting the probability of choosing the first action by agent one and two, respectively. Similarly to Singh et al. (2000); Zhang & Lesser (2010), we analyze the dynamics of $\theta_1$ and $\theta_2$ for LOLA, LA, and naïve agents to investigate the coordination behaviors.

**Theorem 1** *If, in the previously defined two-person, two-action coordination game with a miscoordination regret $g$, the agents are updated following the LA method and a fixed prediction length $\eta$, then they are subject to miscoordination for $g > 1/2\eta$. If the agents are updated following the LOLA method and a fixed prediction length $\eta$, then they are subject to miscoordination for $g > 1/4\eta$. If the agents follow the naïve updates, then they are never subject to miscoordination for any value of g.*
**Proof –** See Appendix C.1.

A closer inspection of the HOG methods reveals two important aspects about their fundamental ideas. First, anticipating other agents' learning is only effective when it is close to their true future learning. Existing HOG methods assume a reasoning level for other agents. If this assumption is wrong, it can negatively affect the coordination among the cooperative agents. Second, the idea of shaping other agents' learning can be misleading if the other agents do not follow, making agents more likely to suffer from miscoordination. We hypothesize that by addressing these two aspects of current HOG methods, miscoordination among agents can be avoided.

### 5.2 HIERARCHICAL REASONING

Based on our hypothesis, we propose *Hierarchical Reasoning* (HR), an HOG methodology especially designed for cooperative agents. In contrast to standard HOG methods, HR determines a hierarchy among the agents in each training iteration, which determines the reasoning orders of the agents. Concretely, if $n = 2$, and we assume that the first agent is the *leader* and the second agent is the *follower*, the gradient adjustment for the leader is similar to first-order LOLA agents, and is:

$$\nabla_{\theta_1} V^{\text{Leader}}(s, \theta_1, \theta_2 + \Delta\theta_2) \approx \nabla_{\theta_1} V + (\nabla_{\theta_2 \theta_1} V)^\intercal \Delta\theta_2 + (\nabla_{\theta_1} \Delta\theta_2)^\intercal \nabla_{\theta_2} V, \quad (8)$$

where $V = V(s, \theta_1, \theta_2)$ is the common value function, and $\Delta\theta_2 = \eta \nabla_{\theta_2} V$. However, unlike LOLA agents, the leader knows the reasoning level of the follower, which is a naïve agent. The plan for the leader is to change its parameters $\bar{\theta}_1 = \theta_1 + \nabla_{\theta_1} V^{\text{Leader}}(s, \theta_1, \theta_2 + \Delta\theta_2)$ in such a way that an optimal increase in the common value is achieved, after its new parameters are taken into account by the follower. Therefore, the follower must follow the plan and adjust its parameters through

$$\nabla_{\theta_2} V^{\text{Follower}}(s, \bar{\theta}_1, \theta_2) \approx \nabla_{\theta_2} V + (\nabla_{\theta_1 \theta_2} V)^\intercal \nabla_{\theta_1} V^{\text{Leader}}(s, \theta_1, \theta_2 + \Delta\theta_2), \quad (9)$$

**Theorem 2** *If, in the previously defined two-person, two-action coordination game with a miscoordination regret $g$, the agents are updated following the HR methodology, then they are not subject to miscoordination for any value of g.*
**Proof –** See Appendix C.2.

With this, we have shown that HR naturally avoids miscoordination and therefore, our hypothesis is correct. However, the main goal is to demonstrate that HR improves coordination among common-interested agents with respect to naïve learning, which does not take into account higher-order gradients at all. If HR does not improve the coordination, there is no clear benefit over the naïve learners,

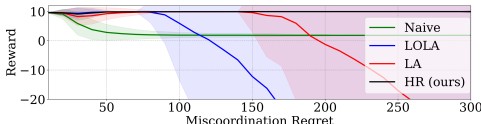

Figure 4: Converged results for various values of miscoordination regret.

as they also avoid miscoordination. To show the benefits of HR, we employ a standard two-agent, three-action coordination game (Claus & Boutilier, 1998). The game has a common reward matrix $R = \begin{bmatrix} 10 & 0 & k \\ 0 & 2 & 0 \\ k & 0 & 10 \end{bmatrix}$, and we define $g = 10 - k$ as the miscoordination regret. Each agent is parameterized with three parameters: $\theta^1$, $\theta^2$, and $\theta^3$ ($\theta^i > 0 \ \forall i \in \{1, 2, 3\}$ and $\sum_i^3 \theta^i = 1$), representing

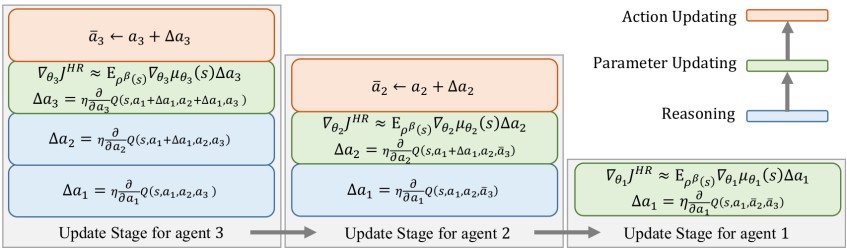

Figure 5: An example of the parameter update stages in HR-MADDPG, for a game with three common-interested agents $m = 3$, assigned to three hierarchy levels.

the probability of taking the actions $a^1$, $a^2$, and $a^3$, respectively. The game has two global equilibrium points (if $\theta^1 = 1$ or $\theta^3 = 1$ for all agents), and one local equilibrium point (if $\theta^2 = 1$ for all agents). In Figure 4, we depict the converged results for this game for naïve, LA, LOLA, and HR agents, for various values of miscoordination regret $g$. The experiments are run 500 times until convergence, with random initializations. From 4, we find that both LA and LOLA agents are subject to miscoordination for high values of $g$, which is consistent with our findings for the two-action coordination game. However, the most interesting aspect of this experiment is that by increasing the value of $g$, the coordination among the naïve agents reduces, leading them to the local equilibrium point, whereas our HR agents consistently achieve the highest reward, independently of the miscoordination regret. This shows the benefit of HR agents over naïve agents.

### 5.2.1 HR-MADDPG

In this section, we propose HR-MADDPG, an extension of HOG-MADDPG for games with common-interested agents and high-dimensional state spaces. We first define $\mathcal{M} \subseteq \mathcal{N}$, as a set of common-interested agents such that $R_i = R_j \ \forall i, j \in \mathcal{M}$. Without the loss of generality, we consider $\mathcal{M} = \mathcal{N}$, i.e., team games with a common state-action value function $Q(s, a_1, ..., a_m)$.

**Hierarchy level assignment.** In the policy update step of HR-MADDPG, each agent is first assigned to one of $m = |\mathcal{M}|$ levels of hierarchy based on the amount of influence that it has on other agents, i.e., its shaping capacity, in each training iteration. The shaping capacity of the $i^{\text{th}}$ agent, $f_i$, is the sum of $l^2$-norms of the shaping term in Eq. 1 with respect to all other agents $j$:

$$f_i = \sum_{j \neq i \ \& \ \in \mathcal{M}} \left\| (\nabla_{a_i} \Delta a_j)^\intercal \nabla_{a_j} Q(a_1, ..., a_m) \right\|, \tag{10}$$

where $\Delta a_j = \nabla_{a_j} Q(a_1, ..., a_m)$. In each hierarchy level, the assigned agent is a leader of the lower hierarchy levels and a follower of the higher ones, with two reasoning rules: 1) a leader knows the reasoning levels of the followers and is one level higher, and 2) a follower cannot reason about the leaders and only follows their shaping plans. As HR-MADDPG benefits from centralized learning, the only constraint for these reasoning rules remains the centralized state-action value function.

**Parameter update.** After the hierarchy level assignment, the agents update their policy parameters in $m$ update stages, i.e., one for each agent, and in a top-down fashion: the agent in the highest hierarchy level updates its policy parameters first. In each update stage, the corresponding agent 1) reasons about the followers (if any) in a bottom-up fashion, i.e., it reasons about the agent in the lowest hierarchy level first, 2) updates its policy parameters, and 3) updates its action for the next update stage (if any). Figure 5 demonstrates an example with the update stages for three common-interested agents 1, 2 and 3, that are assigned to hierarchy levels $h^1$, $h^2$, $h^3$, where agent 3, assigned to $h^3$ is the leader, etc. For the case of $m$ agents, see the HR-MADDPG update rules in Alg. 3.

### 5.3 EXPERIMENTS

In this section, we aim to demonstrate the advantages of our proposed HR-MADDPG for games with common-interested agents, compared to 1) LA-MADDPG and LOLA-MADDPG, and 2) state-of-the-art methods: MADDPG (Lowe et al., 2017), CPG-MADDPG (Peng et al., 2021), and PR2-MADDPG (Wen et al., 2019). For this purpose, we first develop the Particle Coordination game to assess the coordination capability of the methods carefully. Then, we compare the general performance of all the methods in standard multi-agent games (Lowe et al., 2017; Peng et al., 2021). See Appendix D.3 and D.4 for details on the experiments.

| Methods | ↑NAER in Particle Environment | | | ↑NAER in Mujoco Environment | | |
|---|---|---|---|---|---|---|
| | Cooperative Navigation | Physical Deception | Predator-Prey | Half-Cheetah | Walker | Reacher |
| DDPG (LB) | 0.00 | 0.00 | 0.00 | 0.00 | 0.00 | 0.00 |
| C-MADDPG (UB) | 1.00 | 1.00 | 1.00 | 1.00 | 1.00 | 1.00 |
| MADDPG | 0.77 | 0.61 | 0.21 | 0.86 | 0.45 | 0.02 |
| CPG-MADDPG | 0.78 | 0.67 | 0.18 | 0.88 | 0.46 | 0.05 |
| PR2-MADDPG | 0.78 | 0.54 | 0.08 | 0.85 | 0.45 | 0.01 |
| LA-MADDPG (ours) | 0.78 | 0.63 | 0.13 | 0.85 | 0.43 | 0.04 |
| LOLA-MADDPG (ours) | 0.77 | 0.56 | 0.13 | 0.83 | 0.42 | 0.01 |
| HR-MADDPG (ours) | **0.88** | **0.83** | **0.44** | **0.94** | **0.67** | **0.42** |

Table 3: Comparisons of methods in terms of the Normalized Average Episode Reward (NAER) for common-interested agents. LB: Lower Bound. UB: Upper Bound.

### 5.3.1 PARTICLE COORDINATION GAME

Our proposed game is a variant of the Cooperative Navigation game (Lowe et al., 2017) with two agents and three landmarks. The agents should select and approach one of the landmarks, and the landmark closest to an agent is considered to be the selected landmark. If the agents select and approach the same landmark, they receive global or local optimal rewards based on the selected landmark. They will receive an assigned miscoordination penalty if they select and approach different landmarks. In Figure 6, we depict the learning curves for our method and other MADDPG-based algorithms. From this figure, it is clear that our HR-MADDPG is the only method that consistently converges to the global optimum of the game, which is consistent with our previous results coordination games. Further experiments regarding the sensitivity of HOG-MADDPG methods to the prediction length are provided in Appendix D.3.

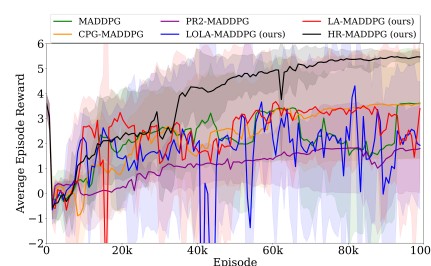

Figure 6: Learning curves in the Particle Coordination game. Higher values are better.

### 5.3.2 STANDARD MULTI-AGENT GAMES

We evaluate the methods in three Particle environment games (Lowe et al., 2017): 1) Cooperative Navigation with three common-interested agents, 2) Physical Deception with two common-interested and one self-interested agent, and 3) Predator-Prey with two common-interested (predator) and one self-interested (prey) agents. Furthermore, we compare the methods in three games within the multi-agent Mujoco environment (Peng et al., 2021): 1) two-agent Half-Cheetah, 2) two-agent Walker, and 3) two-agent Reacher. In the mixed environments (Physical Deception and Predator-Prey), we have employed the MADDPG method for the self-interested agents. We report the Normalized Average Episode Reward for the common-interested agents in Table 3, where the normalization is done between the single-agent variant of MADDPG (DDPG (Lillicrap et al., 2016)) and a fully centralized (in learning and execution) variant of MADDPG, referred to as C-MADDPG. In Table 3, we observe that our proposed HOG-MADDPG consistently and significantly outperforms all the state-of-the-art MADDPG-based methods. Again, these results confirm that our proposed HR-MADDPG improves coordination among common-interested agents, leading to better results.

## 6 DISCUSSION

In this paper, we proposed the HOG-MADDPG framework to make HOG methods applicable to games with high-dimensional state spaces. As a result, the benefits of HOG can now be used in many more MARL problems. As a first case study, we investigated the applicability of current HOG methodologies to team games, and found that they suffer from miscoordination, which we then solved with our proposed HR methodology. Like this solution, there are numerous other possibilities for extending HOG-MADDPG, e.g., factorizing the centralized value functions, which is essential for many-agent games. With our work, we aim to spark such new ideas for HOG methods in MARL, and we provide the framework to realize them.

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

## APPENDIX

## A MORE DETAILS ON THE HOG-MADDPG METHODS

### A.1 LA-MADDPG

In the standard MADDPG, the performance objective is defined as:

$$J_i = \mathbb{E}_{\rho^\beta(s,\hat{a})} Q_i(s, \hat{a}_1, ..., a_i, ..., \hat{a}_n, \theta_1, ..., \theta_n)|_{a_i=\mu_{\theta_i}(s)}, \tag{11}$$

where $\rho^\beta(s,\hat{a})$ is the state-action distribution of the behavior policy with $\hat{a} = \{\hat{a}_i \forall i \in \mathcal{N}\}$. The gradient of $J_i$ with respect to $\theta_i$ can be approximated as:

$$\nabla_{\theta_i} J_i \approx \mathbb{E}_{\rho^\beta(s,\hat{a})} \nabla_{\theta_i} Q_i(s, \hat{a}_1, ..., a_i, ..., \hat{a}_n)|_{a_i=\mu_{\theta_i}(s)}, \tag{12}$$

where the direct dependencies of state-action value function to policy parameters can be dropped based on the proofs presented in Degris et al. (2012). Given $n = 2$ and a naïve opponent for simplicity, the gradient adjustment for parameters of the LA-MADDPG agent ($\theta_1$) is computed by accounting for the anticipated policy parameters of the opponent, i.e., $\hat{\theta}_2 + \Delta\hat{\theta}_2(s)$:

$$\nabla_{\theta_1} J_1^{\text{LA}} \approx \mathbb{E}_{\rho^\beta(s)} \nabla_{\theta_1} Q_1(s, a_1, \tilde{a}_2)|_{a_1=\mu_{\theta_1}(s), \tilde{a}_2=\mu_{\hat{\theta}_2+\Delta\hat{\theta}_2}(s)}, \tag{13}$$

where $\Delta\hat{\theta}_2 = \eta \nabla_{\hat{\theta}_2} Q_2(s, \hat{a}_1, \hat{a}_2)|_{\hat{a}_1=\mu_{\hat{\theta}_1}(s), \hat{a}_2=\mu_{\hat{\theta}_2}(s)}$, and $\hat{\theta}_1$ and $\hat{\theta}_2$ are the behavior policy parameters. As the agents cannot have access to these parameters, we propose to project the anticipated gradient information to the action space using first-order Taylor expansion:

$$\begin{aligned} \tilde{a}_2 &= \mu_{\hat{\theta}_2+\Delta\hat{\theta}_2}(s) \\ &\approx \mu_{\hat{\theta}_2}(s) + (\Delta\hat{\theta}_2)^\intercal \nabla_{\hat{\theta}_2} \mu_{\hat{\theta}_2}(s) \end{aligned} \tag{14}$$

---

**Algorithm 1:** LA-MADDPG for a set of $n$ self-interested agents ($\mathcal{N}$).

---

Initialize $\mu_{\theta_i}$, $Q_i$, $\mu_i'$, and $Q_i'$ $\forall i \in \mathcal{N}$
**for** episode $= 1$ to max-num-episodes **do**
  Receive initial state $s$
  **for** $t = 1$ to max-episode-length **do**
    Select action $a_i$ from $\mu_{\theta_i}(s)$ and the exploration strategy $\forall i \in \mathcal{N}$
    Execute actions $a = \{a_i\}_{\forall i \in \mathcal{N}}$ and observe rewards $r = \{r_i\}_{\forall i \in \mathcal{N}}$ and new state $s'$
    Store the tuple $(s, a, r, s')$ in replay buffer $\mathcal{D}$
    Set $s = s'$

    Sample a random $K$ tuples $\{(s^k, a^k, r^k, s'^k)\}_{k \in \{1,...,K\}}$ from $\mathcal{D}$
    **for** agent $i = 1$ to $n$ **do**
      Set $y_i^k = r_i^k + \gamma Q_i'(s'^k, a_1', ..., a_n')|_{a_h' = \mu_h'(s'^k)}$, for $k \in \{1, ..., K\}$
      Update state-action value function $Q_i$ by minimizing:

$$\mathcal{L} = \frac{1}{K} \sum_{k \in \{1,...,K\}} [(Q_i(s^k, a_1^k, ..., a_n^k) - y_i^k)^2]$$

    **end for**
    **for** agent $i = 1$ to $n$ **do**
      **for** agent $j = 1$ to $n$ **do**
        Project the anticipated gradients
        **if** $j = i$ **then** continue
        Set $\Delta a_j^k = \eta \frac{\partial}{\partial a_j^k} Q_j(s^k, a_1^k, ..., a_n^k)$ for $k \in \{1, ..., K\}$
      **end for**
      Update policy parameters $\theta_i$ via: $\nabla_{\theta_i} J_i^{\text{LA}} \approx$

$$\frac{1}{K} \sum_{k \in \{1,...,K\}} \nabla_{\theta_i} \mu_{\theta_i}(s^k) \frac{\partial}{\partial a_i^k} Q_i(s^k, a_1^k + \Delta a_1^k, ..., a_i^k, ..., a_n^k + \Delta a_n^k)|_{a_i^k = \mu_{\theta_i}(s^k)}$$

    **end for**
    Update $Q_i'$ and $\mu_i'$ $\forall i \in \mathcal{N}$
  **end for**
**end for**

---

Given that:

$$\begin{aligned}
\Delta \hat{\theta}_2 &= \eta \nabla_{\hat{\theta}_2} Q_2(s, \hat{a}_1, \hat{a}_2) \\
&= \eta \nabla_{\hat{\theta}_2} \mu_{\hat{\theta}_2}(s) \left( \nabla_{\hat{a}_2} Q_2(s, \hat{a}_1, \hat{a}_2)|_{\hat{a}_2 = \mu_{\hat{\theta}_2}(s)} \right)^{\mathsf{T}},
\end{aligned} \tag{15}$$

we have:

$$\begin{aligned}
\tilde{a}_2 &\approx \mu_{\hat{\theta}_2}(s) + \left( \eta \nabla_{\hat{\theta}_2} \mu_{\hat{\theta}_2}(s) \left( \nabla_{\hat{a}_2} Q_2(s, \hat{a}_1, \hat{a}_2) \right)^{\mathsf{T}} \right)^{\mathsf{T}} \nabla_{\hat{\theta}_2} \mu_{\hat{\theta}_2}(s) \\
&= \mu_{\hat{\theta}_2}(s) + \nabla_{\hat{a}_2} Q_2(s, \hat{a}_1, \hat{a}_2) \left( \eta \nabla_{\hat{\theta}_2} \mu_{\hat{\theta}_2}(s) \right)^{\mathsf{T}} \nabla_{\hat{\theta}_2} \mu_{\hat{\theta}_2}(s) \\
&= \mu_{\hat{\theta}_2}(s) + \nabla_{\hat{a}_2} Q_2(s, \hat{a}_1, \hat{a}_2) \eta \left\| \nabla_{\hat{\theta}_2} \mu_{\hat{\theta}_2(s)} \right\|^2 \\
&= \mu_{\hat{\theta}_2}(s) + \nabla_{\hat{a}_2} Q_2(s, \hat{a}_1, \hat{a}_2) \hat{\eta},
\end{aligned} \tag{16}$$

where $\|.\|$ is the $l^2$-norm and we have defined the projected prediction length $\hat{\eta} = \eta \left\| \nabla_{\hat{\theta}_2} \mu_{\hat{\theta}_2(s)} \right\|^2$ since $\left\| \nabla_{\hat{\theta}_2} \mu_{\hat{\theta}_2(s)} \right\|^2$ is a positive number and independent of $\theta_1$. Therefore:

$$\begin{aligned}
\tilde{a}_2 &\approx \mu_{\hat{\theta}_2}(s) + \hat{\eta} \nabla_{\hat{a}_2} Q_2(s, \hat{a}_1, \hat{a}_2) \\
&= \hat{a}_2 + \Delta \hat{a}_2.
\end{aligned} \tag{17}$$

---

**Algorithm 2:** LOLA-MADDPG for a set of $n$ self-interested agents ($\mathcal{N}$).

---

Initialize $\mu_{\theta_i}$, $Q_i$, $\mu_i'$, and $Q_i'$ $\forall i \in \mathcal{N}$
**for** episode $= 1$ to max-num-episodes **do**
   Receive initial state $s$
   **for** $t = 1$ to max-episode-length **do**
      Select action $a_i$ from $\mu_{\theta_i}(s)$ and the exploration strategy $\forall i \in \mathcal{N}$
      Execute actions $a = \{a_i\}_{\forall i \in \mathcal{N}}$ and observe rewards $r = \{r_i\}_{\forall i \in \mathcal{N}}$ and new state $s'$
      Store the tuple $(s, a, r, s')$ in replay buffer $\mathcal{D}$
      Set $s = s'$

      Sample a random $K$ tuples $\{(s^k, a^k, r^k, s'^k)\}_{k \in \{1,...,K\}}$ from $\mathcal{D}$
      **for** agent $i = 1$ to $n$ **do**
         Set $y_i^k = r_i^k + \gamma Q_i'(s'^k, a_1', ..., a_n')|_{a_h' = \mu_h'(s'^k)}$, for $k \in \{1, ..., K\}$
         Update state-action value function $Q_i$ by minimizing:

$$\mathcal{L} = \frac{1}{K} \sum_{k \in \{1,...,K\}} [(Q_i(s^k, a_1^k, ..., a_n^k) - y_i^k)^2]$$

      **end for**
      Set $a_i^k = \mu_{\theta_i}(s^k)$, for $k \in \{1, ..., K\}$ and $i \in \mathcal{N}$
      **for** agent $i = 1$ to $n$ **do**
         **for** agent $j = 1$ to $n$ **do**
            Project the anticipated gradients
            **if** $j = i$ **then** continue
            Set $\Delta a_j^k = \eta \frac{\partial}{\partial a_j^k} Q_j(s^k, a_1^k, ..., a_n^k)$ for $k \in \{1, ..., K\}$
         **end for**
         Update policy parameters $\theta_i$ via: $\nabla_{\theta_i} J_i^{\text{LA}} \approx$

$$\frac{1}{K} \sum_{k \in \{1,...,K\}} \nabla_{\theta_i} \mu_{\theta_i}(s^k) \frac{\partial}{\partial a_i^k} Q_i(s^k, a_1^k + \Delta a_1^k, ..., a_i^k, ..., a_n^k + \Delta a_n^k)$$

      **end for**
      Update $Q_i'$ and $\mu_i'$ $\forall i \in \mathcal{N}$
   **end for**
**end for**

---

Replacing Eq. 17 in Eq. 13 yields Eq. 5:

$$\nabla_{\theta_1} J_1^{\text{LA}} \approx \mathbb{E}_{\rho^\beta(s,\hat{a})} \nabla_{\theta_1} \mu_{\theta_1}(s) \nabla_{a_1} Q_1(s, a_1, \hat{a}_2 + \Delta \hat{a}_2)|_{a_1 = \mu_{\theta_1}(s)}. \tag{18}$$

In the case of $n$ agents, the agent $i \in \mathcal{N}$ first anticipates the gradient information of all agents $j \in \mathcal{N}$ in the action space as:

$$\Delta \hat{a}_j = \hat{\eta} \nabla_{\hat{a}_2} Q_2(s, \hat{a}_1, ..., \hat{a}_n) \tag{19}$$

Then, agent $i$ updates its parameters $\theta_i$ by the following gradient adjustment:

$$\nabla_{\theta_i} J_i^{\text{LA}} \approx \mathbb{E}_{\rho^\beta(s,\hat{a})} \nabla_{\theta_i} \mu_{\theta_i}(s) \nabla_{a_i} Q_i(s, \hat{a}_1 + \Delta \hat{a}_1, ..., a_i, ..., \hat{a}_n + \Delta \hat{a}_n)|_{a_i = \mu_{\theta_i}(s)}. \tag{20}$$

Please refer to Alg. 1 for more details on the LA-MADDPG optimization framework.

### A.2 LOLA-MADDPG

In CPG-MADDPG, the gradient update is:

$$\nabla_{\theta_i} J_i \approx \mathbb{E}_{\rho^\beta(s)} \nabla_{\theta_i} Q_i(s, a_1, ..., a_n)|_{a_i = \mu_{\theta_i}(s)} \forall i \in \mathcal{N}. \tag{21}$$

Given $n = 2$ and a naïve opponent for simplicity, the gradient adjustment for parameters of the LOLA-MADDPG agent ($\theta_1$) is computed by accounting for the anticipated policy parameters of the

---

**Algorithm 3:** HR-MADDPG for a set of $m$ common-interested agents ($\mathcal{M}$).

---

Initialize $\mu_{\theta_i}$, $Q_i$, $\mu'_i$, and $Q'_i$ $\forall i \in \mathcal{M}$

**for** episode $= 1$ to max-num-episodes **do**

    Receive initial state $s$

    **for** $t = 1$ to max-episode-length **do**

        Select action $a_i$ from $\mu_{\theta_i}(s)$ and the exploration strategy $\forall i \in \mathcal{M}$

        Execute actions $a = \{a_i\}_{\forall i \in \mathcal{M}}$ and observe common reward $r$ and new state $s'$

        Store the tuple $(s, a, r, s')$ in replay buffer $\mathcal{D}$

        Set $s = s'$

        Sample a random $K$ tuples $\{(s^k, a^k, r^k, s'^k)\}_{k \in \{1,...,K\}}$ from $\mathcal{D}$

        Set $y^k = r^k + \gamma Q'(s'^k, a'_1, ..., a'_m)|_{a'_h = \mu'_h(s'^k)}$, for $k \in \{1, ..., K\}$

        Update state-action value function $Q$ by minimizing:

$$\mathcal{L} = \frac{1}{K} \sum_{k \in \{1,...,K\}} [(Q(s^k, a_1^k, ..., a_m^k) - y^k)^2]$$

        Assign the agents into $m$ hierarchy levels using Eq. 10

        Set $a_i^k = \mu_{\theta_i}(s^k)$, for $k \in \{1, ..., K\}$ and $i \in \mathcal{M}$

        **for** agent $i = m$ to 1 **do**

            **for** agent $j = 1$ to $i$ **do**

                Project the anticipated gradients

                Compute $\Delta a_j^k$, for $k \in \{1, ..., K\}$:

                **if** $j = 1$ & $i \neq m$ **then** $\Delta a_1^k = \eta \frac{\partial}{\partial a_1^k} Q(s^k, a_1^k, ..., a_i^k, \bar{a}_{i+1}^k, ..., \bar{a}_m^k)$

                **elif** $j \neq 1$ & $i = m$ **then** $\Delta a_j^k = \eta \frac{\partial}{\partial a_j^k} Q(s^k, a_1^k + \Delta a_1^k, ..., a_{j-1}^k + \Delta a_{j-1}^k, a_j^k, ..., a_m^k)$

                **elif** $j = 1$ & $i = m$ **then** $\Delta a_1^k = \eta \frac{\partial}{\partial a_1^k} Q(s^k, a_1^k, ..., a_m^k)$

                **else** $\Delta a_j^k = \eta \frac{\partial}{\partial a_j^k} Q(s^k, a_1^k + \Delta a_1^k, ..., a_{j-1}^k + \Delta a_{j-1}^k, a_j^k, \bar{a}_{j+1}^k, ...\bar{a}_m^k)$

            **end for**

            Update policy parameters $\theta_i$ via:

$$\nabla_{\theta_i} J_i^{\text{HR}} \approx \frac{1}{K} \sum_{k \in \{1,...,K\}} \nabla_{\theta_i} \mu_{\theta_1}(s^k) \Delta a_i^k$$

            Set $\bar{a}_i^k = \text{detach}(a_i^k + \Delta a_i^k)$, for $k \in \{1, ..., K\}$

        **end for**

        Update $Q'_i$ and $\mu'_i$ $\forall i \in \mathcal{M}$

    **end for**

**end for**

---

opponent, i.e., $\theta_2 + \Delta\theta_2(s)$:

$$\nabla_{\theta_1} J_1^{\text{LOLA}} \approx \mathbb{E}_{\rho^\beta(s)} \nabla_{\theta_1} Q_1(s, a_1, \tilde{a}_2)|_{a_1 = \mu_{\theta_1}(s), \tilde{a}_2 = \mu_{\theta_2 + \Delta\theta_2}(s)}, \tag{22}$$

where $\Delta\theta_2 = \eta \nabla_{\theta_2} Q_2(s, a_1, a_2)|_{a_1 = \mu_{\theta_1}(s), a_2 = \mu_{\theta_2}(s)}$, and unlike for LA-MADDPG, is a function of $\theta_1$. Again, we propose to project the anticipated gradient information to the action space. Given that:

$$\Delta\theta_2 = \eta \nabla_{\theta_2} \mu_{\theta_2}(s) \left( \nabla_{a_2} Q_2(s, a_1, a_2)|_{a_1 = \mu_{\theta_1}(s), a_2 = \mu_{\theta_2}(s)} \right)^\mathsf{T}, \tag{23}$$

we have:

$$\begin{aligned}
\tilde{a}_2 &= \mu_{\theta_2 + \Delta\theta_2}(s) \\
&\approx \mu_{\theta_2}(s) + (\Delta\theta_2)^\mathsf{T} \nabla_{\theta_2} \mu_{\theta_2}(s) \\
&= \mu_{\theta_2}(s) + \nabla_{a_2} Q_2(s, a_1, a_2) \eta \left\| \nabla_{\theta_2} \mu_{\theta_2(s)} \right\|^2 \\
&= \mu_{\theta_2}(s) + \nabla_{a_2} Q_2(s, a_1, a_2) \hat{\eta},
\end{aligned} \tag{24}$$

where $\|.\|$ is the $l^2$-norm and we have defined the projected prediction length $\hat{\eta} = \eta \left\| \nabla_{\theta_2} \mu_{\theta_2(s)} \right\|^2$ since $\left\| \nabla_{\theta_2} \mu_{\theta_2(s)} \right\|^2$ is a positive number and independent of $\theta_1$. Therefore:

$$\tilde{a}_2 \approx a_2 + \Delta a_2. \tag{25}$$

Replacing Eq. 25 in Eq. 22 yields Eq. 7:

$$\nabla_{\theta_1} J_1^{\text{LOLA}} \approx \mathbb{E}_{\rho^\beta(s)} \nabla_{\theta_1} \mu_{\theta_1}(s) \nabla_{a_1} Q_1(s, a_1, a_2 + \Delta a_2)\big|_{a_1=\mu_{\theta_1}(s), a_1=\mu_{\theta_2}(s)}. \tag{26}$$

In the case of $n$ agents, the agent $i \in \mathcal{N}$ first anticipates the gradient information of all agents $j \in \mathcal{N}$ in the action space as:

$$\Delta a_j = \hat{\eta} \nabla_{a_2} Q_2(s, a_1, ..., a_n) \tag{27}$$

Then, agent $i$ updates its parameters $\theta_i$ by the following gradient adjustment:

$$\nabla_{\theta_i} J_i^{\text{LA}} \approx \mathbb{E}_{\rho^\beta(s,a)} \nabla_{\theta_i} \mu_{\theta_i}(s) \nabla_{a_i} Q_i(s, a_1 + \Delta a_1, ..., a_i, ..., a_n + \Delta a_n)\big|_{a_i=\mu_{\theta_i}(s)}. \tag{28}$$

Please refer to Alg. 2 for more details on the LOLA-MADDPG optimization framework.

### A.3 HR-MADDPG

For the general cases of $m$ agents, we provide the HR-MADDPG algorithm in Alg. 3.

## B THEORETICAL ANALYSES ON THE PROJECTION ESTIMATION

### B.1 INFLUENCE OF THE PROJECTION ESTIMATION

As discussed in Appendices A.1 and A.2, we project the anticipated gradient information to the action space using first-order Taylor expansion. This section studies the influence of this projection estimation on the performance of the HOG methods. Without the loss of generality, we consider LOLA-MADDPG with two agents (naïve opponents). Using first-order Taylor expansion, we showed in Appendix A.2 that the anticipated gradient of the second agent on the parameter space is projected to the action space as:

$$\mu_{\theta_2 + \Delta\theta_2}(s) \approx \mu_{\theta_2}(s) + \hat{\eta}_{\text{1st}} \nabla_{a_2} Q_2(s, a_1, a_2), \tag{29}$$

where $\Delta\theta_2 = \eta \nabla_{\theta_2} Q_2(s, a_1, a_2)$, $\eta \in \mathbb{R}^+$ is the prediction length, and we denote $\hat{\eta}_{\text{1st}} \in \mathbb{R}^+$ as the projected prediction length via first-order Taylor expansion, which is obtained as (see Appendix A.2):

$$\hat{\eta}_{\text{1st}} = \eta \left\| \nabla_{\theta_2} \mu_{\theta_2(s)} \right\|^2, \tag{30}$$

**Theorem 3** *If the anticipated gradients are projected using first-order Taylor expansion, for sufficiently small $\hat{\eta}_{1st}$, there exists $\eta' \in \mathbb{R}^+$ such that*

$$\mu_{\theta_2 + \Delta\theta'_2}(s) = \mu_{\theta_2}(s) + \hat{\eta}_{1st} \nabla_{a_2} Q_2(s, a_1, a_2), \tag{31}$$

*where*

$$\begin{aligned} \Delta\theta'_2 &= \eta' \nabla_{\theta_2} Q_2(s, a_1, a_2) \\ \hat{\eta}_{1st} &= \eta \left\| \nabla_{\theta_2} \mu_{\theta_2(s)} \right\|^2 \\ \eta &\in \mathbb{R}^+ \end{aligned} \tag{32}$$

In order to prove Theorem 3, we first need to show that:

**Lemma 1** *If the anticipated gradients are projected using full-order Taylor expansion, there exists $\hat{\eta}_{full} \in \mathbb{R}$ such that*

$$\mu_{\theta_2 + \Delta\theta_2}(s) = \mu_{\theta_2}(s) + \hat{\eta}_{full} \nabla_{a_2} Q_2(s, a_1, a_2), \tag{33}$$

*where*

$$\begin{aligned} \Delta\theta_2 &= \eta \nabla_{\theta_2} Q_2(s, a_1, a_2) \\ \eta &\in \mathbb{R}^+ \end{aligned} \tag{34}$$

**Proof –** The full-order Taylor expansion of the anticipated gradient yields:

$$\mu_{\theta_2+\Delta\theta_2}(s) = \mu_{\theta_2}(s) + (\Delta\theta_2)^{\mathsf{T}}\nabla_{\theta_2}\mu_{\theta_2}(s) + \frac{1}{2}(\Delta\theta_2)^{\mathsf{T}}H_{\mu_{\theta_2}}(s)\Delta\theta_2 + O(\|\Delta\theta_2\|^3), \qquad (35)$$

where $H_{\mu_{\theta_2}}(s)$ denotes the Hessian of $\mu_{\theta_2}$ at $s$. Given that:

$$
\begin{aligned}
\Delta\theta_2 &= \eta\nabla_{\theta_2}Q_2(s, a_1, a_2) \\
&= \eta\nabla_{\theta_2}\mu_{\theta_2}(s)\left(\nabla_{a_2}Q_2(s, a_1, a_2)\right)^{\mathsf{T}},
\end{aligned}
\qquad (36)
$$

we have

$$
\begin{aligned}
\mu_{\theta_2+\Delta\theta_2}(s) =&\mu_{\theta_2}(s) + \nabla_{a_2}Q_2(s, a_1, a_2)\eta\left\|\nabla_{\theta_2}\mu_{\theta_2(s)}\right\|^2 \\
&+ \frac{1}{2}\nabla_{a_2}Q_2(s, a_1, a_2)\eta^2\left(\nabla_{\theta_2}\mu_{\theta_2}(s)\right)^{\mathsf{T}}H_{\mu_{\theta_2}}(s)\nabla_{\theta_2}\mu_{\theta_2}(s)\left(\nabla_{a_2}Q_2(s, a_1, a_2)\right)^{\mathsf{T}} \\
&+ O(\eta^3).
\end{aligned}
\qquad (37)
$$

By defining

$$
\begin{aligned}
C1(s) &= \left\|\nabla_{\theta_2}\mu_{\theta_2(s)}\right\|^2 \\
C_2(s) &= \frac{1}{2}\left(\nabla_{\theta_2}\mu_{\theta_2}(s)\right)^{\mathsf{T}}H_{\mu_{\theta_2}}(s)\nabla_{\theta_2}\mu_{\theta_2}(s)\left(\nabla_{a_2}Q_2(s, a_1, a_2)\right)^{\mathsf{T}},
\end{aligned}
\qquad (38)
$$

we have:

$$\mu_{\theta_2+\Delta\theta_2}(s) =\mu_{\theta_2}(s) + \nabla_{a_2}Q_2(s, a_1, a_2)(\eta C_1(s) + \eta^2 C_2(s) + O(\eta^3)), \qquad (39)$$

Given the definition of $C_2(s)$ and the dimension constraint implied by $\nabla_{a_2}Q_2(s, a_1, a_2)$, it can be concluded that $C_1(s) \in \mathbb{R}^+$ and $C_2(s) \in \mathbb{R}$. Therefore:

$$\mu_{\theta_2+\Delta\theta_2}(s) =\mu_{\theta_2}(s) + \hat{\eta}_{\text{full}}\nabla_{a_2}Q_2(s, a_1, a_2), \qquad (40)$$

where $\hat{\eta}_{\text{full}} = \eta C_1(s) + \eta^2 C_2(s) + O(\eta^3) \in \mathbb{R}$. Consequently, we have proved Lemma 1

If we now project the anticipated gradients, with a prediction length $\eta' \in \mathbb{R}^+$, to the action space using full-order Taylor expansion, we have:

$$\mu_{\theta_2+\Delta\theta'_2}(s) = \mu_{\theta_2}(s) + \hat{\eta}'_{\text{full}}\nabla_{a_2}Q_2(s, a_1, a_2), \qquad (41)$$

where $\hat{\eta}'_{\text{full}} = \eta'C_1(s) + \eta'^2 C_2(s) + O(\eta'^3)$. In order to prove Theorem 3, we need to find the values of $\hat{\eta}_{\text{1st}}$ that yields:

$$
\begin{aligned}
\hat{\eta}_{\text{1st}} &= \hat{\eta}'_{\text{full}} \\
&= \eta'C_1(s) + \eta'^2 C_2(s) + O(\eta'^3),
\end{aligned}
\qquad (42)
$$

and at the same time $\eta' \in \mathbb{R}^+$. By neglecting $O(\eta'^3)$ and given that $\hat{\eta}_{\text{1st}} \in \mathbb{R}^+$, there are two cases to be considered:

- if $C_2(s)$ is non-negative, then for any value of $\hat{\eta}_{\text{1st}} \in \mathbb{R}^+$, there exists $\eta \in \mathbb{R}^+$.
- if $C_2(s)$ is negative, then for $\hat{\eta}_{\text{1st}} < \frac{C_1(s)^2}{4|C_2(s)|}$, there exists $\eta \in \mathbb{R}^+$.

Therefore, for sufficiently small $\hat{\eta}_{\text{1st}}$, i.e., $\hat{\eta}_{\text{1st}} < \frac{C_1(s)^2}{4|C_2(s)|}$, there always exists $\eta \in \mathbb{R}^+$, and consequently, the Theorem 3 is proved.

Based on Theorem 3, the projection estimation via first-order Taylor expansion and sufficiently small $\hat{\eta}_{\text{1st}}$ only scales the prediction length as both $\eta$ and $\eta'$ are non-negative numbers. The amount of this scaling depends on both values of $C_1(s)$ and $C_2(s)$.

The general theoretical analyses on single-sate games reveal that scaling the prediction length directly influences the HOG methods' convergence speed (Letcher et al., 2019; Zhang & Lesser, 2010). However, by directly changing the projected prediction length, i.e., $\hat{\eta}_{\text{1st}}$, we can tune the resulting prediction length in the state space, i.e., $\eta'$, and consequently improve the convergence behavior. To empirically show this, we conducted an experimental study to analyze the influence of $\hat{\eta}_{\text{1st}}$ on the convergence behavior of LOLA-MADDPG in the Exit-room game (See Figure 7). The experiments are repeated four times, and the mean results are reported in the form of normalized average episode reward in Figure 7. It is clear from Figure 7 that increasing $\hat{\eta}_{\text{1st}}$ improves the convergence behavior of LOLA-MADDPG. However, high values of the projected prediction length ($\hat{\eta}_{\text{1st}} = 1.3$ in Figure 7) can lead to instability of the algorithm which can be attributed to our findings in Theorem 3.

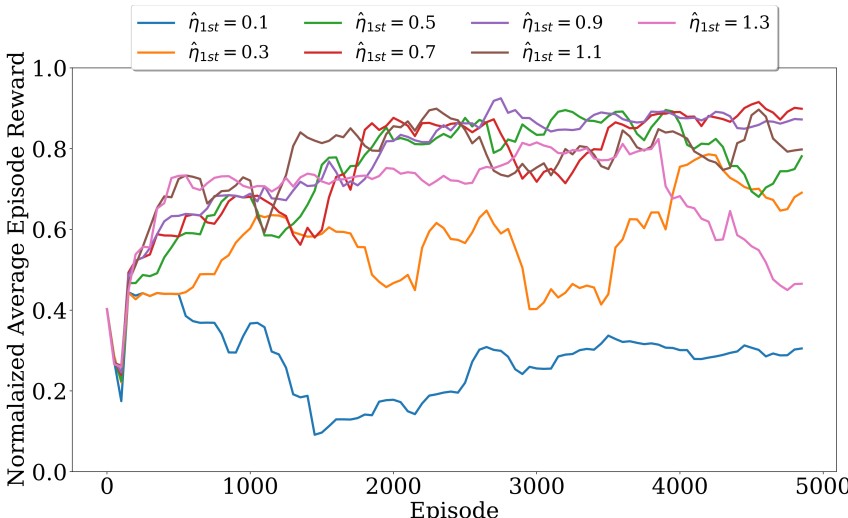

Figure 7: The influence of $\hat{\eta}_{1st}$ on the convergence behavior of LOLA-MADDPG in the Exit-room game.

### B.2 TIME COMPLEXITY OF THE PROJECTION ESTIMATION

In this section, we first study the time complexity of gradient anticipation in the parameter space. Then, we discuss the time complexity reduction we gain by projecting the anticipated gradients to the action space.

As both policy and state-action value functions are approximated via neural networks, the time complexity of the gradient anticipation follows the time complexity of backpropagation in neural networks. Without the loss of the generality, we assume (as done in our experiments) that policy and state-action value networks have the same number of hidden layers, $H$, and neurons in each hidden layer, $N$. Therefore, the backpropagation time complexity of the networks for an input state of size $N_s$ and action of size $N_a$ is (Lister & Stone, 1995):

- Backpropagation time complexity in the policy network: $O(N_s N + (H-1)N^2 + NN_a)$
- Backpropagation time complexity in the state-action value network: $O((N_s + N_a)N + (H-1)N^2 + N)$

Given that $N > N_s + N_a$, the time complexity of both networks can be upper bounded by $O(LN^2)$ where we defined $L = H + 1$. As discussed in Appendix A.2, the anticipated gradient in the state space for the case of the two-agent LOLA-MADDPG is:

$$\Delta\theta_2 = \eta\nabla_{\theta_2}\mu_{\theta_2}(s)\left(\nabla_{a_2}Q_2(s,a_1,a_2)|_{a_1=\mu_{\theta_1}(s),a_2=\mu_{\theta_2}(s)}\right)^\top. \tag{43}$$

Therefore, the time complexity of gradient anticipation in the state space is $O(LN^2) \times O(LN^2)$, or in other words, $O(L^2N^4)$. This is while the projected anticipated gradient in the action space is:

$$\Delta a_2 = \hat{\eta}\nabla_{a_2}Q_2(s,a_1,a_2), \tag{44}$$

which has the complexity of $O(LN^2)$. Consequently, by projecting the anticipated gradient to the action space, the time complexity is reduced by $O(LN^2)$.

## C   MORE DETAILS ON THE MISCOORDINATION ANALYSES

The two-agent, two-action coordination game (Claus & Boutilier, 1998) is defined by a common reward matrix $R = \left[\begin{smallmatrix} a & k \\ k & a \end{smallmatrix}\right]$, where $a > 0$ is coordination reward and $k \leq 0$ is the miscoordination penalty. We further define $g = a - k > 0$ as the miscoordination regret. Let $\theta_1 \in [0,1]$ and $\theta_2 \in [0,1]$ denote the probability of choosing the first action by first and second agents, respectively.

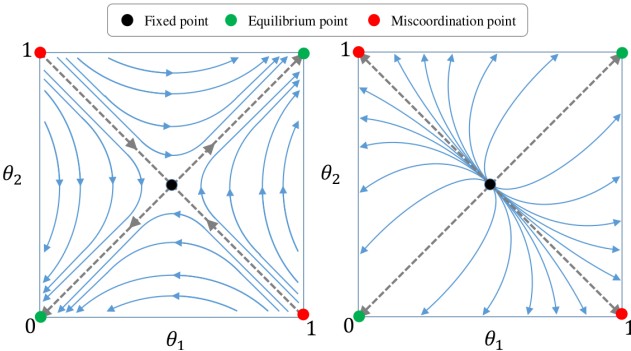

Figure 8: Phase planes of the LA and LOLA dynamics. Left: unstable fixed point. Right: unstable saddle fixed point.

The common value function $V(\theta_1, \theta_2) = 2g(\theta_1\theta_2) - g(\theta_1 + \theta_2) + a$ is the expected reward, given $\theta_1$ and $\theta_2$. The game has two equivalent Nash equilibria: $\theta_1 = \theta_2 = 0$ and $\theta_1 = \theta_2 = 1$. We further define two miscoordination points: $(\theta_1 = 1, \theta_2 = 0)$ and

### C.1 Proof of Theorem 1

Given Eq. 2 the unconstrained dynamics of LA agents can be defined by the following differential equations :

$$\begin{bmatrix} \partial\theta_1/\partial t \\ \partial\theta_2/\partial t \end{bmatrix} = \begin{bmatrix} 4\eta g^2 & 2g \\ 2g & 4\eta g^2 \end{bmatrix} \begin{bmatrix} \theta_1 \\ \theta_2 \end{bmatrix} - \begin{bmatrix} 2\eta g^2 + g \\ 2\eta g^2 + g \end{bmatrix} \tag{45}$$

This system of equations has a unique fixed point (zero gradients) at $\theta_1 = \theta_2 = 0.5$ (see Figure 8). The eigenvalue analysis of the coefficient matrix yields two real eigenvalues, $\lambda_1 = 4\eta g^2 + 2g$ and $\lambda_2 = 4\eta g^2 - 2g$, and two respective diagonal and off-diagonal eigenvectors. While $\lambda_1$ is always positive, the sign of $\lambda_2$ depends on the values of both $\eta$ and $g$. For a fixed prediction length, non-positive values of $\lambda_2$ are reached by $g \leq \frac{1}{2\eta}$. In this case, the fixed point is an unstable saddle point (or unstable line in case of $\lambda_2 = 0$), and the agents, with any initial values of $\theta_1$ and $\theta_2$ (except on the fixed point itself), converge to the equilibrium points (see Figure 8-Left). However, when the miscoordination regret increases, $g > \frac{1}{2\eta}$, the fixed point becomes an unstable (source) point (see Figure 8-Right). Therefore, some initial values of $\theta_1$ and $\theta_2$ naturally lead to the miscoordination points $(\theta_1 = 0, \theta_2 = 1)$.

In the case of LoLA agents, the unconstrained dynamics can be defined as:

$$\begin{bmatrix} \partial\theta_1/\partial t \\ \partial\theta_2/\partial t \end{bmatrix} = \begin{bmatrix} 8\eta g^2 & 2g \\ 2g & 8\eta g^2 \end{bmatrix} \begin{bmatrix} \theta_1 \\ \theta_2 \end{bmatrix} - \begin{bmatrix} 4\eta g^2 + g \\ 4\eta g^2 + g \end{bmatrix} \tag{46}$$

This system of equations has a unique fixed point, again at $\theta_1 = \theta_2 = 0.5$ (see Figure 8). The eigenvalue analysis of the coefficient matrix yields two real eigenvalues, $\lambda_1 = 8\eta g^2 + 2g$ and $\lambda_2 = 8\eta g^2 - 2g$, and two respective diagonal and off-diagonal eigenvectors. Similar to the case of LA agents, $\lambda_1$ is always positive and the sign of $\lambda_2$ depends on the values of both $\eta$ and $g$. For a fixed prediction length, non-positive values of $\lambda_2$ are reached by $g \leq \frac{1}{4\eta}$. In this case, the fixed point is an unstable saddle point (or unstable line in case of $\lambda_2 = 0$), and the agents, with any initial values of $\theta_1$ and $\theta_2$ (except on the fixed point itself), converge to the equilibrium points. However, when the miscoordination regret increases, $g > \frac{1}{4\eta}$, the fixed point becomes an unstable (source) point. Therefore, some initial values of $\theta_1$ and $\theta_2$ naturally lead to the miscoordination points.

In the case of the naïve agents, we have:

$$\begin{bmatrix} \partial\theta_1/\partial t \\ \partial\theta_2/\partial t \end{bmatrix} = \begin{bmatrix} 0 & 2g \\ 2g & 0 \end{bmatrix} \begin{bmatrix} \theta_1 \\ \theta_2 \end{bmatrix} - \begin{bmatrix} g \\ g \end{bmatrix} \tag{47}$$

Similar to the case of LOLA and LA, this system of equations has a unique fixed point (zero gradients) at $\theta_1 = \theta_2 = 0.5$. The eigenvalue analysis of the coefficient matrix yields two real eigenvalues, $\lambda_1 = 2g$ and $\lambda_2 = -2g$, and two respective diagonal and off-diagonal eigenvectors. This time, however, the eigenvalues are of opposite signs for any values of $g$ and the fixed point is always an

unstable saddle point. Therefore, any initial values of $\theta_1$ and $\theta_2$ (except on the fixed point itself) naturally lead to the equilibrium points.

Based on these results, we hypothesize that 1) wrong reasoning level assumptions and 2) shaping plans that are not followed can lead to miscoordination points. Both LOLA and LA agents assume that other agents are naïve learners, which is obviously a wrong assumption since all agents conduct first-order reasoning. For self-interested agents, it is natural for the agents to don't unveil their reasoning order to each other. as they have different goals. However, common-interested agents can benefit more from this reasoning information to achieve their common goal. Furthermore, LOLA agents constantly underestimate other LOLA agents and try to shape them. This is while other LOLA agents do not follow the plan, and each tries to show that it is smarter than the others. Letcher et al. (2019) shows that these arrogant behaviors lead to outcomes that are strictly worse for everyone. It is also clear from Theorem 1 that the range of $g$ that leads to miscoordination in LOLA ($g > \frac{1}{4\eta}$) is larger than the range of $g$ in LA ($g > \frac{1}{2\eta}$).

In the above coordination game, one general solution to reduce the possibility of miscoordination is decreasing the prediction length's values. However, in non-tabular settings, with large state spaces, it is infeasible to estimate the miscoordination regret and adjust the prediction length accordingly. Furthermore, the prediction length directly affects the usage of higher-order gradient information, and further reducing the prediction length ($\eta \to 0$) leads to the naïve learners.

### C.2 PROOF OF THEOREM 2

Given Eqs. 9 and 8, the unconstrained dynamics of the HR agents can be defined by the following differential equations :

$$
\begin{bmatrix} \partial\theta_1/\partial t \\ \partial\theta_2/\partial t \end{bmatrix} = \begin{bmatrix} 8\eta g^2 & 2g \\ 2g + 16\eta^2 g^3 & 4\eta g^2 \end{bmatrix} \begin{bmatrix} \theta_1 \\ \theta_2 \end{bmatrix} - \begin{bmatrix} 4\eta g^2 + g \\ 8\eta^2 g^3 + 4\eta g^2 + g \end{bmatrix} \tag{48}
$$

resulting in a unique fixed point at $\theta_1 = \theta_2 = 0.5$ and two real eigenvalues, $\lambda = 6\eta g^2 \pm 2p\sqrt{9eta^2 g^2 + 1}$. Unlike the case of LOLA and LA, the eigenvalues are now of opposite signs for any values of $g$, and the fixed point is always an unstable saddle point. Therefore, any initial values of $\theta_1$ and $\theta_2$ (except on the fixed point itself) naturally lead to the equilibrium points.

## D   MORE DETAILS ON THE EXPERIMENTS AND THE IMPLEMENTATIONS

In this section, we describe all the experiments and the implementations in detail. To ease the reproducibility of our work, the code of our methods and experiments are shared with the community at [to comply with the double-blind policy, the link will be inserted in the final version].

**A note on partial observability**. So far, we have formulated the MARL setup as an MG, where it is assumed that the agents have access to the state space. However, in many games, the agents only receive a private state observation of the current state. In this case, the MARL setup can be formulated as a Partially Observable Markov Game (PO-MG) (Littman, 1994). A PO-MG is a tuple $(\mathcal{N}, \mathcal{S}, \{\mathcal{A}_i\}_{i \in \mathcal{N}}, \{\mathcal{O}_i\}_{i \in \mathcal{N}}, \{\mathcal{R}_i\}_{i \in \mathcal{N}}, \mathcal{T}, \{\Omega_i\}_{i \in \mathcal{N}}, \rho, \gamma)$, where $\mathcal{O}_i$ is the set of sate observations for agent $i \in \mathcal{N}$. Each agent $i$ chooses its action $a_i \in \mathcal{A}_i$ through the policy $\pi_{\theta_i} : \mathcal{O}_i \times \mathcal{A}_i \to [0, 1]$ parameterized by $\theta_i$ conditioning on the given state observation $o_i \in \mathcal{O}_i$. After transition to a new state, each agent $i$ receives a private state observation through its observation function $\Omega_i : \mathcal{S} \to \mathcal{O}_i$. In this case, the centralized state-action value function for each agent $i$ is defined as $Q_i(o_1, ..., o_n, a_1, ..., a_n) = \mathbb{E}[G_i^t(\tau | s^t = s, o_i = \Omega_i(s) \ \& \ a_i^t = a_i \forall i \in \mathcal{N})]$. Therefore, the proposed HOG-MADDPG framework can be modified accordingly.

### D.1   MATRIX GAMES

Iterated Rotational Game (IRG)(Zhang & Lesser, 2010) is a one-state, two-agent, one-action matrix game with the reward matrices depicted in Table 4 (for two discrete actions). Each agent must choose a 1-D continuous action ($a_1$ for agent one and $a_2$ for agent two) representing the probability of taking two discrete actions. The game has a unique equilibrium point at $a_1 = a_2 = 0.5$, which is also the fixed point of the game. IRG is originally proposed to demonstrate the circular behavior that

|  | discrete action 1 | discrete action 2 |
|---|---|---|
| discrete action 1 | $(0, 3)$ | $(3, 2)$ |
| discrete action 2 | $(1, 0)$ | $(2, 1)$ |

Table 4: Reward matrix in IRG.

|  | Cooperate | Defect |
|---|---|---|
| Cooperate | $(-1, -1)$ | $(-3, 0)$ |
| Defect | $(0, -3)$ | $(-3, -3)$ |

Table 5: Reward matrix in IPD.

can emerge if the agents follow the naïve gradient updates. LA agents, on the other hand, can quickly converge to the equilibrium point by considering their opponent's parameter adjustment. As LOLA agents cannot preserve the fixed point of the game (Letcher et al., 2019), they converge to non-equilibrium points. We evaluate the performances of methods based on the Distance to Equilibrium (DtE).

Iterated Prisoner's Dilemma (IPD) (Foerster et al., 2018a) is a five-state, two-agent, two-action game with the reward matrices depicted in Table 5. Each agent must choose between two discrete actions (cooperate or defect). The game is played for 150 time steps ($T = 150$). In the one-shot version of the game, there is only one Nash equilibrium for the agents (Defect, Defect). In the iterated games, (Defect, Defect) is also a Nash equilibrium. However, a better equilibrium is Tit-For-Tat (TFT), where the players start by cooperating and then repeat the previous action of the opponents. The LOLA agents can shape the opponent's learning to encourage cooperation and, therefore, converge to TFT (Letcher et al., 2019). We evaluate the methods' performances based on the Averaged Episode Reward (AER).

**Implementation details**. We used policies and state-action value functions with the same neural network architecture in all methods. We employed Multi-Layer Perceptron (MLP) networks with two hidden layers of dimension 64 for policies and state-action value functions. In order to make the value functions any-order differentiable, we used SiLU nonlinear function (Elfwing et al., 2018) in between the hidden layers. For IRG, we used the Sigmoid function in the policies to output 1-D continues action, and for IPD, we used the Gumble-softmax function (Jang et al., 2017) in the policies to output two discrete actions. The algorithms are trained for 900 (in IRG) and 50 (in IPD) episodes by running Adam optimizer (Kingma & Ba, 2015) with a fixed learning rate of $0.01$. The prediction length, $\eta$, for LOLA and LA agents in both HOG-MADDPG and DiCE frameworks are fixed to 1 in all experiments. We reported the best methods' performances in Table 1. All experiments are repeated five times, and the results are reported in terms of mean and standard deviation in Figure 1 and Table 1.

## D.2 EXIT-ROOM GAME

The Exit-Room game is a grid-world variant of the IPD, with two agents (blue and red) and $15^{2l}$ states where $l \in \{1, 2, 3\}$ is the complexity level of the game. The agents should cooperate and move towards the exit doors on the right. However, they are tempted to exit the left doors and, in some cases, not exiting at all. In level 1, the agents have three possible actions (*move-left*, *move-right*, or *do nothing*), and the reward is computed as Vinitsky et al. (2019):

$$\text{reward}_C = \lambda_C(\text{cooperation}_{self} + \text{cooperation}_{opponent})$$
$$\text{reward}_D = \lambda_D(1 - \text{cooperation}_{self}) \qquad (49)$$
$$\text{reward} = \text{reward}_C + \text{reward}_D,$$

where $\lambda_C$ and $\lambda_D$ are some constants, and $\text{cooperation}_{self}$ and $\text{cooperation}_{opponent}$ are the normalized distances of the agent and its opponent to the right door, respectively. In levels 2 and 3, the agents have additional *move-up* and *move-down* actions. In level 3, the door positions are randomly located, resulting in more complex interactions among the agents. In addition to the reward in Eq. (49), the agents receive an additional reward for approaching the doors in levels 2 and 3. Each agent receives four $90 \times 90$ RGB images representing the state observations of the last four time steps.

**Implementation details**. As before, we used policies and state-action value functions with the same neural network architecture in all methods. Both policy and value networks consist of two parts: encoder and decoder. The encoders are CNN networks with three convolutional layers ($12 \times 90 \times 90 \rightarrow 32 \times 21 \times 21 \rightarrow 64 \times 9 \times 9 \rightarrow 64 \times 7 \times 7$ ) and two fully connected layers ($3136 \rightarrow 512 \rightarrow 128$ ), with SiLU nonlinear functions (Elfwing et al., 2018) in between. The decoders are MLP

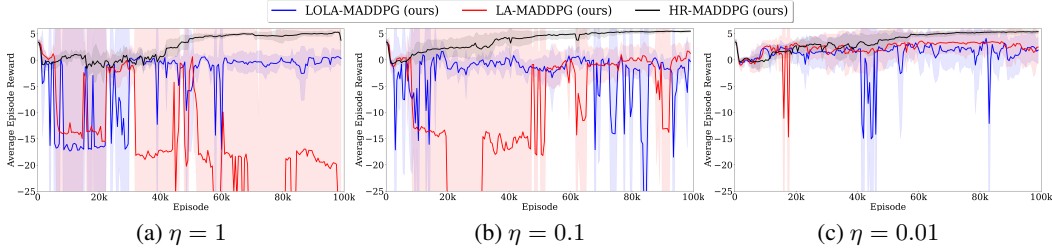

Figure 9: Learning curves in the particle coordination game with different values of the prediction length, $\eta$, for HOG-MADDPG methods.

networks with two hidden layers of dimension 64 for policies and state-action value functions. We used the Gumble-softmax function in the policies (Jang et al., 2017) to output the discrete actions. The algorithms are trained for 450 (in level one) and 4500 (in levels two and three) episodes by running Adam optimizer (Kingma & Ba, 2015) with a fixed learning rate of $0.01$. The prediction length, $\eta$, for LOLA and LA agents in both HOG-MADDPG and DiCE frameworks are fixed to 1 in all experiments. All experiments are repeated five times, and the results are reported in terms of mean and standard deviation in Figure 2 and Table 2. The methods are evaluated in terms of the Normalized Average Episode Reward (NAER), where the normalization is done between the highest and lowest episode rewards in each game level. We reported the best methods' performances in Table 2.

### D.3 PARTICLE COORDINATION GAME

Our proposed game is a variant of the Cooperative Navigation game in the Particle environment (Lowe et al., 2017). As shown in Figure 10, each one of the two agents (purple circles) should select and approach one of the three landmarks (one gray and two green circles). Landmarks are selected based on the closest distance between the agent and the landmarks. Suppose the agents select and approach the same landmark. In that case, they receive global (by selecting the green landmarks) or local (by selecting the gray landmark) optimal rewards. They will receive an assigned miscoordination penalty if they select and approach different landmarks. Each agents receive a 14-D state observation vector and select a 5-D, one-hot vector, representing one of the five discrete ac-

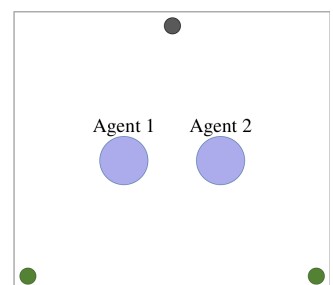

Figure 10: Particle coordination game with two agents and three landmarks.

tions: *move-right*, *move-left*, *move-up*, *move-down*, and *stay*. The horizon is set to 25, $T = 25$. The agents receive a Landmark Selection (LS) reward indicated by the matrix: $R_{LS} = \begin{bmatrix} 2 & 0 & -20 \\ 0 & 0.4 & 0 \\ -20 & 0 & 2 \end{bmatrix}$, where the rows and columns indicate the selected landmarks by the first and second agents, respectively. Furthermore, the agents receive an additional reward for approaching the landmarks.

**Implementation details**. We used policies and state-action value functions with the same neural network architecture in all methods. We employed MLP networks with two hidden layers of dimension 64 for policies and state-action value functions with SiLU nonlinear functions (Elfwing et al., 2018) in between. We used the Gumble-softmax function (Jang et al., 2017) in the policies to output the discrete actions. The algorithms are trained for 100k episodes by running Adam optimizer (Kingma & Ba, 2015) with a fixed learning rate of $0.01$. We set the prediction length, $\eta$, to $0.1$ for HR agents and $0.01$ for LOLA and LA agents. All experiments are repeated five times, and the results are reported in terms of mean and standard deviation in Figure 6.

**Effect of the prediction length**. Additionally, we illustrate the effect of various prediction lengths on the performance of HR, LOLA, and LA agents in Figure 9. As shown, the HR-MADDPG method consistently converges to the global optimum of the game. This is while both LOLA-MADDPG and LA-MADDPG demonstrate weak performances, especially by increasing the prediction length.

| | Particle Environment | | | Mujoco Environment | | |
|---|---|---|---|---|---|---|
| | Cooperative Navigation | Physical Deception | Predator-Prey | Half-Cheetah | Walker | Reacher |
| Observation | 18-D | 10-D (8-D) | 14-D (12-D) | 11-D | 11-D | 8-D |
| Observation type | continuous | continuous | continuous | continuous | continuous | continuous |
| # Discrete observations | infinite | infinite | infinite | infinite | infinite | infinite |
| Action | 5-D | 5-D (5-D) | 5-D (5-D) | 3-D | 3-D | 1-D |
| Action type | discrete | discrete | discrete | continuous | continuous | continuous |
| Policy parameter | 11K-D | 11K-D (8-D) | 10K-D (12-D) | 70K-D | 70K-D | 70K-D |
| Horizon (step) | 25 | 25 | 25 | 100 | 300 | 50 |

Table 6: Specifications in the standard multi-agent games. In the mixed environments, the dimensions are reported as "$d_1$ $(d_2)$" where $d_1$ is the dimension for common-interested agents and $d_2$ is the dimension for self-interested ones.

## D.4 STANDARD MULTI-AGENT GAMES

We evaluate the methods in three Particle environment games (Lowe et al., 2017): 1) Cooperative Navigation with three common-interested agents, 2) Physical Deception with two common-interested and one self-interested agent, and 3) Predator-Prey with two common-interested (predator) and one self-interested (prey) agents. Furthermore, we compare the methods in three games within the multi-agent Mujoco environment (Peng et al., 2021): 1) two-agent Half-Cheetah, 2) two-agent Walker, and 3) two-agent Reacher. In the mixed environments (Physical Deception and Predator-Prey), we have employed the MADDPG method for the self-interested agents. Games' specifications are reported in Table 6. We created separate validation and test sets for each game that included 100 and 300 randomly generated scenarios, respectively. In each game, we save the models that have the best performance on the validation set and test them on the test set to report the results. All experiments are repeated five times, and the mean results are reported in Table 3 in terms of the Normalized Average Episode Reward (NAER). The normalization is done between the single-agent variant of MADDPG (DDPG (Lillicrap et al., 2016)) and a fully centralized (in learning and execution) variant of MADDPG, referred to as C-MADDPG. The non-normalized data are reported in Table 8 in terms of the Collective Average Episode Reward (CAER) for the common-interested agents.

**Implementation details**. As before, we used policies and state-action value functions with the same neural network architecture in all methods. We employed MLP networks with two hidden layers (of dimension 64 for the Particle environment and 256 for the Mujoco environment) for policies and state-action value functions with SiLU nonlinear functions (Elfwing et al., 2018). In the Particle environment, We used the Gumble-softmax function (Jang et al., 2017) in the policies to output the discrete actions and trained the algorithms for 100k episodes by running Adam optimizer (Kingma & Ba, 2015) with a fixed learning rate of $0.01$. In the Mujoco environment, we used the Tanh function in the policies to output the continuous actions and train the algorithms for 10k episodes by running Adam optimizer (Kingma & Ba, 2015) with a fixed learning rate of $0.001$. The prediction lengths, $\eta$, in HR-MADDPG, LA-MADDPG, and LOLA-MADDPG are optimized between $0.001 - 0.1$ all games. We avoid considering any smaller value than $0.001$ for the prediction length as it makes the algorithms similar to the naïve learners. The optimized prediction lengths are reported in Table 7.

**Ablation study**. We have additionally conducted an ablation study on the hierarchy level assignments in the HR-MADDPG. Rather than iteratively sorting the agents based on their shaping capacities through Eq. (10), we randomly assigned the agents to hierarchy levels in the beginning and fixed the hierarchy levels throughout the optimization. This variant of the HR-MADDPG, referred to as HR-MADDPG (F), is evaluated and compared in Table 9. As can be seen, using the proposed sorting strategy based on the shaping capacities of the agents, as done in our HR-MADDPG, constantly improves performance.

| | $\eta$ in Particle Environment | | | $\eta$ in Mujoco Environment | | |
|---|---|---|---|---|---|---|
| | Cooperative Navigation | Physical Deception | Predator-Prey | Half-Cheetah | Walker | Reacher |
| LA-MADDPG | 0.002 | 0.003 | 0.01 | 0.001 | 0.002 | 0.003 |
| LOLA-MADDPG | 0.001 | 0.003 | 0.008 | 0.001 | 0.001 | 0.002 |
| HR-MADDPG | 0.003 | 0.01 | 0.04 | 0.004 | 0.004 | 0.007 |

Table 7: The optimized prediction lengths for HOG-MADDPG methods.

| | ↑CAER in Particle Environment | | | ↑CAER in Mujoco Environment | | |
|---|---|---|---|---|---|---|
| Methods | Cooperative Navigation | Physical Deception | Predator-Prey | Half-Cheetah | Walker | Reacher |
| DDPG (LB) | -189.18 | 13.04 | 10.20 | 611.74 | 2925.17 | -16.19 |
| C-MADDPG (UB) | -130.91 | 20.82 | 20.87 | 1564.60 | 6362.36 | -8.89 |
| MADDPG | -144.02 | 17.82 | 12.46 | 1435.93 | 4482.31 | -16.02 |
| CPG-MADDPG | -143.96 | 18.29 | 12.08 | 1454.62 | 4511.08 | -15.84 |
| PR2-MADDPG | -143.67 | 17.26 | 11.08 | 1418.69 | 4473.14 | -16.09 |
| LA-MADDPG | -143.72 | 17.92 | 11.58 | 1424.39 | 4408.56 | -15.89 |
| LOLA-MADDPG | -144.21 | 17.43 | 11.54 | 1406.61 | 4369.13 | -16.13 |
| HR-MADDPG | **-137.63** | **19.48** | **14.85** | **1503.07** | **5226.26** | **-13.16** |

Table 8: Comparisons of methods in terms of the Collective Average Episode Reward (CAER) for common-interested agents, corresponding to the normalized data in Table 3.

| | ↑NAER in Particle Environment | | | ↑NAER in Mujoco Environment | | |
|---|---|---|---|---|---|---|
| Methods | Cooperative Navigation | Physical Deception | Predator-Prey | Half-Cheetah | Walker | Reacher |
| HR-MADDPG (F) | 0.85 | 0.80 | 0.38 | 0.92 | 0.63 | 0.38 |
| HR-MADDPG | **0.88** | **0.83** | **0.44** | **0.94** | **0.67** | **0.42** |

Table 9: Ablation study on the hierarchy level assignments in our HR-MADDPG method.

