# OpenReview forum: "Revisiting Higher-Order Gradient Methods for Multi-Agent Reinforcement Learning"
_ICLR.cc/2023/Conference — Submitted to ICLR 2023_

### Official Review · Reviewer_TWCf · 2022-10-12

**Confidence:** 2
**Correctness:** 4
**Technical Novelty And Significance:** 3
**Empirical Novelty And Significance:** 2
**Recommendation:** 5

**Clarity, Quality, Novelty And Reproducibility:**

The paper is clear and the experiments look sound to me. I indeed have some concerns regarding novelty as mentioned above.

**Strength And Weaknesses:**

Pros:
- The LOLA and LA approximations makes preserving high-order gradient more efficient and the improvement is shown via experiment results
- The hierarchical reasoning approach further improves the performance in fully-cooperative case.

Cons:
- The two approximation scheme, LA and LOLA, are existing approaches well known in the community. So how do you place your novelty using these methods for approximation purposes?
- The term high-oerder gradient is actually very confusing. At the very beginning I was thinking this is use higher-order derivatives to do policy optimization. It would be super helpful if the author make a comment on this early in the intro part.

**Summary Of The Paper:**

The paper revisits the idea of using higher-order gradient in multi-agent RL with two improvements: (1) using LOLA and LA for preserving higher-order gradients and (2) use a hierarchical reasoning approach to coordinate agents in team (cooperation) scenario.

**Summary Of The Review:**

The paper revisits the idea of using higher-order gradient in multi-agent RL with two improvements: (1) using LOLA and LA for preserving higher-order gradients and (2) use a hierarchical reasoning approach to coordinate agents in team (cooperation) scenarios. So far I am not fully convinced of the novelty of the paper.

---

> ### Author Response · Authors · 2022-11-14
> **Response to Reviewer TWCf [3/3]**
>
> > 3. The term high-oerder gradient is actually very confusing. At the very beginning I was thinking this is use higher-order derivatives to do policy optimization. It would be super helpful if the author make a comment on this early in the intro part.
>
>
> We thank the Reviewer for this comment, and we would like to clear up this misunderstanding. As a matter of fact, High-Order Gradient (HOG) methods do use higher-order derivatives to do policy optimization. In the HOG methods, the agents anticipate the learning step of other agents. As agents use gradient-based optimization, learning anticipation naturally leads to the usage of higher-order gradients. To better clarify this, we investigate various cases in LOLA with one agent (policy is parameterized by $\theta_1$) and one opponent (policy is parameterized by $\theta_2$).
>
>  1. If the agent is a naive learner, then the policy gradient adjustment for the agent is the standard first-order gradient of the value function:
>
> $\nabla_{\theta_1}V_1(s,\theta_1,\theta_2)$
>
>  2. If the agent does first-order reasoning, it assumes the opponent to be a naive learner. Then, the policy gradient adjustment for the agent includes second-order gradients:
>
> $\nabla_{\theta_1}V_1^{\text{1-LOLA}}(s,\theta_1,\theta_2+\eta\nabla_{\theta_2}V_2(s,\theta_1,\theta_2))=\nabla_{\theta_1}V_1(s,\theta_1,\theta_2) +...+ {f\Bigl( \nabla_{\theta_2\theta_1}V_2(s,\theta_1,\theta_2)\Bigr)}$
>
>  3. If the agent does second-order reasoning, it assumes the opponent is doing first-order reasoning. Then, the policy gradient adjustment for the agent includes third-order gradients:
>
> $\nabla_{\theta_1}V_1^{\text{2-LOLA}}(s,\theta_1,\theta_2+\eta\nabla_{\theta_2}V_2(s,\theta_1+\eta\nabla_{\theta_1}V_1(s,\theta_1,\theta_2),\theta_2))=\nabla_{\theta_1}V_1(s,\theta_1,\theta_2) +...+  {f\Bigl(\nabla_{\theta_1\theta_2\theta_1}V_1(s,\theta_1,\theta_2)\Bigr)}$
>
>
> We would like to thank the Reviewer again for taking the time to review our manuscript. We kindly ask the Reviewer to let us know if the responses are clear, or if the Reviewer desires any further clarifications.
>
>     [1] Alistair Letcher, Jakob Foerster, David Balduzzi, Tim Rockt ̈aschel, and Shimon Whiteson. Stable Opponent Shaping in Differentiable Games. In  International Conference on Learning Representations, 2019.
>     [2] Chongjie Zhang and Victor Lesser. Multi-agent learning with policy prediction. In  Proceedings of the AAAI Conference on Artificial Intelligence, volume 24, 2010.
>     [3] Jakob Foerster, Richard Y Chen, Maruan Al-Shedivat, Shimon Whiteson, Pieter Abbeel, and Igor Mordatch. Learning with Opponent-Learning Awareness. In Proceedings of the 17th International Conference on Autonomous Agents and MultiAgent Systems, pp. 122–130, 2018a.

---

> > ### Comment · Reviewer_TWCf · 2022-11-14
> > **Clarifications**
> >
> > Thanks for the authors' detailed comment. Now I think I understand the other part. My only concern is still the novelty: so can I say that the main contribution you make is to develop new methods to overcome the computation and preservation limitation?

---

> > > ### Author Response · Authors · 2022-11-15
> > > **Response to Reviewer TWCf [1/1]**
> > >
> > > > Thanks for the authors' detailed comment. Now I think I understand the other part. My only concern is still the novelty: so can I say that the main contribution you make is to develop new methods to overcome the computation and preservation limitation?
> > >
> > >
> > > We thank the Reviewer for taking the time to review our responses. The contribution mentioned by the Reviewer, i.e., overcoming the computation and preservation limitations of current HOG methods, is the first novelty we introduce, and is our first main contribution. This contribution lays the groundwork for the second novelty in our work, i.e., improving coordination in team games. Below, we summarize technical, theoretical, and empirical contributions for both of these two high-level novelties.
> > >
> > > ## First Novelty: Overcoming the computation and preservation limitations of current HOG methods
> > >
> > > The applicability of current HOG methods is limited to games with low-dimensional state spaces. We argued that this limitation is due to inefficient computation and preservation of higher-order gradient information, and we propose our HOG-MADDPG framework to solve it.
> > >
> > > ### Technical contributions:
> > >
> > >  - **HOG-MADDPG framework:** We proposed the HOG-MADDPG framework, where we computed higher-order gradients more efficiently than existing methods by optimizing differentiable objective functions, and preserved these gradients more efficiently by projecting these gradients to the action space. (Section 4.2).
> > >  - **LA-MADDPG method:** We developed LA-MADDPG, which applies the LA principles to our HOG-MADDPG framework to make LA applicable to games with high-dimensional state spaces (Section 4.2.1).
> > >  - **LOLA-MADDPG method:** We developed LOLA-MADDPG, which applies the LOLA principles to our HOG-MADDPG framework to make LOLA applicable to games with high-dimensional state spaces (Section 4.2.2).
> > >
> > >
> > > ### Theoretical contributions:
> > >
> > >  - **Validity of projection:** We proved the condition under which the gradient projection is valid, i.e., it does not change the direction of the anticipated gradients in the state space (Theorem 3).
> > >  - **Time complexity reduction:** We proved that our proposed projection concept reduces the time complexity of gradient anticipation by $O(L N^2)$, where $L$ and $N$ are the numbers of layers and neurons per layer in a fully connected policy network, respectively (Appendix B.2).
> > >
> > > ### Empirical contributions:
> > >
> > >  - **Performance boost:** We first developed the Exit-room game with three complexity levels, and then demonstrated that our proposed LOLA-MADDPG significantly outperforms the state-of-the-art HOG method LOLA-DiCE, and that the performance increase is higher with increased complexity (Table 2).
> > >  - **Efficiency boost:** We demonstrated that LOLA-MADDPG is around 130% faster than LOLA-DiCE in terms of the normalized training time per iteration (Table 2).
> > >
> > > ### Take-away message:
> > >
> > >  - We made the existing HOG methods applicable to games with high-dimensional state spaces. Apart from LOLA and LA, many other HOG methods, such as SOS, CO, and SGA (see Section 2), can now be applied to high-dimensional games.
> > >  - Having the HOG-MADDPG framework, we can now use the benefits of HOG in many more MARL problems, e.g., coordination in team games.
> > >
> > > ## Second novelty: Improving coordination in team games
> > >
> > > Existing HOG methods are proposed to improve cooperation in games with self-interested agents, and it is unclear how they perform when agents are fully cooperative, i.e., in team games. In this work, we explore this game setting, find that standard approaches can lead to miscoordination, and propose a method to solve this.
> > >
> > > ### Technical contributions:
> > >
> > >  - **HR:** We proposed HR, a HOG methodology that improves coordination among agents in team games (Section 5.2).
> > >  - **HR-MADDPG method:** We developed HR-MADDPG, which applies the HR principles to our HOG-MADDPG framework to make HR applicable to games with high-dimensional state spaces (Section 5.2.1).
> > >
> > > ### Theoretical contributions:
> > >
> > >  - **Misscoordination in HOG:** We proved that existing HOG methodologies are subject to miscoordination in a matrix coordination game (Theorem 1).
> > >  - **Equilibrium convergence in HR:** We proved that our proposed HR methodology converges to equilibrium in a matrix coordination game (Theorem 2).
> > >
> > > ### Empirical contributions:
> > >
> > >  - **Theorem verification:** We first developed the Particle coordination game and then verified Theorems 1 and 2 in this high-dimensional game (Figure 6).
> > >  - **State of the art:** We show that HR-MADDPG outperforms the state-of-the-art MARL methods in well-known multi-agent games (Table 3).
> > >
> > > ### Take-away message:
> > >
> > >  - To the best of our knowledge, this is the first time that the principles of HOG methods are successfully used in team games, to improve coordination among agents.

---

> ### Author Response · Authors · 2022-11-14
> **Response to Reviewer TWCf [2/3]**
>
> > 2. The two approximation scheme, LA and LOLA, are existing approaches well known in the community. So how do you place your novelty using these methods for approximation purposes?
>
> We thank the Reviewer for this comment. LOLA and LA anticipate the learning step of the opponents for policy optimization by using higher-order gradients. This anticipation is shown to be beneficial for improving coordination among the agents [1]. However, LOLA and LA are only applied to games with low-dimensional games, e.g., matrix games [1],[2],[3]. In our paper, we argued that the reason they cannot be efficiently applied to high-dimensional games is the computation and preservation of high-order gradients:
>
>  - **Computation limitation:** LOLA and LA are currently implemented in the stochastic policy gradient framework and optimize non-differentiable objectives. They require multiple rounds of sampling to compute higher-order gradients. This is very inefficient for high-order reasoning and games with higher-dimensional state spaces, i.e., beyond matrix games.
>  - **Preservation limitation:** The higher-order gradients are computed and preserved in the parameter spaces. When the dimensionality of state spaces increases, e.g., having images as input, the dimensionality of the parameter spaces increases as well. Therefore, computing and storing the higher-order gradient information in high-dimensional parameter spaces is inefficient.
>
> In order to solve these limitations, we proposed the HOG-MADDPG, which is our main novelty.
>
>  - **Overcoming the computation limitation:** In HOG-MADDPG, agents optimize differentiable objectives. Therefore, any-order derivatives can be efficiently computed without sampling.
>  - **Overcoming the preservation limitation:** In HOG-MADDPG, the anticipated gradient information is projected from the policies' parameter spaces to the action spaces. This way, the efficiency is improved as the action spaces have significantly lower dimensionality than the policies' parameter spaces.
>
> Having the HOG-MADDPG framework, we can now use the principles of existing HOG methodologies, e.g., LA and LOLA, for improving coordination in games with higher-dimensional state spaces. For this purpose, we developed LA-MADDPG and LOLA-MADDPG -- applying the LA and LOLA principles, respectively -- which more efficiently compute and preserve the higher-order gradients and are applicable to games with higher-dimensional state spaces.

---

> ### Author Response · Authors · 2022-11-14
> **Response to Reviewer TWCf [1/3]**
>
> We want to thank the Reviewer for the constructive feedback. Below, we individually address all of the Reviewer's concerns and questions.
>
> > 1. The paper revisits the idea of using higher-order gradient in multi-agent RL with two improvements: (1) using LOLA and LA for preserving higher-order gradients and (2) use a hierarchical reasoning approach to coordinate agents in team (cooperation) scenarios. So far I am not fully convinced of the novelty of the paper.
>
> We thank the Reviewer for the comments. We agree with the Reviewer that the novelties of the paper could be specified more explicitly. Therefore, we added a list of contributions in Section 1 of the revised manuscript, and we will briefly explain them here.
>
> Mainly, it appears that there are some misunderstandings regarding the use of LOLA and LA methodologies. We highlight that we are not using LOLA and LA for preserving higher-order gradients. LOLA and LA are existing HOG methods that anticipate the learning step of the opponents for policy optimization by using higher-order gradients, which are computed and preserved in specific manners. Currently, they are only applied to games with low-dimensional state spaces, such as matrix games. The reason is that they inefficiently compute and preserve the higher-order gradients. Therefore, LA and LOLA introduce the problem that we aim to solve in this work, and they are not the solutions, so we do not use them for preserving higher-order gradients. Our proposed solution, called HOG-MADDPG, is a framework to make LA, LOLA, and other HOG methodologies applicable to games with higher-dimensional state spaces by solving the limitations in computation and preservation of higher-order gradients.  Given the proposed framework, we develop two novel methods:
>
>  1. LA-MADDPG, which applies the LA principle but more efficiently
>     computes and preserves the higher-order gradients.
>  2. LOLA-MADDPG, which applies the LOLA principle but again more
>     efficiently computes and preserves the higher-order gradients.
>
> The rest of the contributions are as follows.
> - We demonstrate theoretically, in a two-agent two-action coordination game, and empirically, in a two-agent three-action coordination game, that the existing HOG methodologies can suffer from miscoordination among common-interested agents. To solve this, we propose the HR methodology and show, theoretically and empirically, that it overcomes miscoordination in the coordination games.
>  - We apply the HR principle to our HOG-MADDPG framework and develop HR-MADDPG, a HOG method for common-interested agents. We show that HR-MADDPG outperforms the existing state-of-the-art methods on standard multi-agent games.

---

### Official Review · Reviewer_eo4T · 2022-10-23

**Confidence:** 3
**Clarity, Quality, Novelty And Reproducibility:** The code to reproduce the numerical r…
**Correctness:** 4
**Technical Novelty And Significance:** 2
**Empirical Novelty And Significance:** 2
**Recommendation:** 5

**Strength And Weaknesses:**


Strength:
1.	Multi-Agent Deep Deterministic Policy Gradient (MADDPG)  and projecting gradient from state space to action space are adopted to addressed the issue of computation and preservation of high-order gradient information, respectively.
2.	Hierarchical reasoning is proposed tackle the miscoordination of between common-interested agents.
Weaknesses:
Lack of theoretical analysis on the proposed methods.


**Summary Of The Paper:**


This paper revisits high order gradient methods for multi-agent reinforcement learning and points out that it usually does not apply to games with high-dimensional state spaces due to inefficient computation and preservation of high-order gradient information. The authors develop a scheme to address the problem and enable order gradient methods to work for high dimensional problem. Furthermore, hierarchical reasoning is proposed to improve the coordination in team games.


**Summary Of The Review:**

1.	The review does not think that resorting to MADDPG can totally mitigate the computation difficulty of high-dimensional gradient. Even though  the high order gradient can be calculated, it takes long time when the gradient is of really high-dimensional.
2.	About projecting the gradient information from state space to action space, there is no theoretical analysis on how the projection influence the results. Does it degrade the performance or not? If not, what conditions are supposed to be satisfied.
3.	For the hierarchical reasoning, is it guaranteed to converge to the equilibria for any level of hierarchy.

---

> ### Author Response · Authors · 2022-11-14
> **Response to Reviewer eo4T [3/3]**
>
> > 3. For the hierarchical reasoning, is it guaranteed to converge to the equilibria for any level of hierarchy.
>
> We thank the Reviewer for this comment. In our paper, we have provided the theoretical analyses of Hierarchical Reasoning (HR) for the case of two-agent two-action matrix games. In this situation, the maximum level of the hierarchy is two, and we proved the convergence to equilibrium via eigenvalue analysis of the dynamics of the strategy pairs. For the general cases of $n$ agents (with a maximum of $n$ hierarchy levels), eigenvalue analysis is no longer tractable, and we cannot guarantee convergence. However, we empirically showed in several multiagent games that HR achieves a better performance than existing multi-agent methods.
>
> In [4], the authors provided some theoretical guarantees for the LA methodology in $n$-player differential games by investigating the definiteness of system matrices. Although HR imposes additional complexities to the problem, researching the definiteness of system matrices is a promising future direction for convergence analysis in various hierarchy levels, and could provide more insights into our proposed HR method.
>
> We would like to thank the Reviewer again for taking the time to review our manuscript. We kindly ask the Reviewer to let us know if the responses are clear, or if the Reviewer desires any further clarifications.
>
>     [1] Rashid, T.; Samvelyan, M.; Schroeder, C.; Farquhar, G.; Foerster, J.; Whiteson, S. QMIX: Monotonic Value Function Factorisation for Deep Multi-Agent Reinforcement Learning. In Proceedings of the 35th International Conference on Machine Learning, PMLR, Stockholm, Sweden, 10–15 July 2018; Volume 80, pp. 4295–4304.
>     [2] Foerster, J.; Farquhar, G.; Afouras, T.; Nardelli, N.; Whiteson, S. Counterfactual Multi-Agent Policy Gradients. In Proceedings of the AAAI 2018, Thirty-Second AAAI Conference on Artificial Intelligence, New Orleans, LA, USA, 2–7 February 2018
>     [3] Chongjie Zhang and Victor Lesser. Multi-agent learning with policy prediction. In  Proceedings of the AAAI Conference on Artificial Intelligence, volume 24, 2010.
>     [4] Alistair Letcher, Jakob Foerster, David Balduzzi, Tim Rockt ̈aschel, and Shimon Whiteson. Stable Opponent Shaping in Differentiable Games. In  International Conference on Learning Representations, 2019.

---

> ### Author Response · Authors · 2022-11-14
> **Response to Reviewer eo4T [2/3]**
>
> > 2. About projecting the gradient information from state space to action space, there is no theoretical analysis on how the projection influence the results. Does it degrade the performance or not? If not, what conditions are supposed to be satisfied.
>
>
> We thank the Reviewer for this critical comment. We agree with the Reviewer that our paper needs additional theoretical analyses regarding the gradient projection. We provided a new section, i.e., Appendix B.1, in the revised manuscript to address the theoretical analyses on the influence of the projection estimation. We show in Appendix B.1 that the projection estimation leads to scaling of the prediction length and, consequently, influences the convergence speed of the HOG methods. However, we show that this influence can be compensated by directly tuning the projected prediction length. Our main finding is Theorem 3, and we briefly explain it here.
>
> Without the loss of generality, we consider LOLA-MADDPG with two agents (naive opponents). Using first-order Taylor expansion, we showed in Appendix A.2 that the anticipated gradient of the second agent on the parameter space is projected to the action space as:
>
> $\mu_{{\theta}\_2+\Delta{\theta}\_2}(s) \approx \mu_{{\theta}\_2}(s) + \hat{\eta}\_{\text{1st}}\nabla_{a_2}Q_2(s,a_1,a_2)$
>
> where $\Delta\theta_2 =\eta \nabla_{\theta_2}Q_2(s,a_1,a_2)$, $\eta\in \mathbb{R}^+$ is the prediction length, and we denote $\hat{\eta}_{\text{1st}}\in \mathbb{R}^+$ as the projected prediction length via first-order Taylor expansion, which is obtained as (see Appendix A.2)
>
> $\hat{\eta}\_{\text{1st}}=\eta ||{\nabla_{\theta_2}\mu_{\theta_2(s)}}||^2$
>
>
>
> **Theorem 3:** If the anticipated gradients are projected using first-order Taylor expansion, for sufficiently small $\hat{\eta}_{\text{1st}}$, there exists $\eta'\in \mathbb{R}^+$ such that
>
> $\mu_{{\theta}\_2+\Delta{\theta'}\_2}(s) = \mu_{{\theta}\_2}(s) + \hat{\eta}\_{\text{1st}}\nabla_{a_2}Q_2(s,a_1,a_2)$
>
> where $\Delta\theta'\_2 =\eta' \nabla_{\theta_2}Q_2(s,a_1,a_2)$, $\hat{\eta}\_{\text{1st}}=\eta||{\nabla_{\theta_2}\mu_{\theta_2(s)}}||^2$, and $\eta \in \mathbb{R}^+$
>
> **Proof --** See Appendix B.1.
>
> In Theorem 3, the critera is $\hat{\eta}\_{\text{1st}} < \frac{C_1(s)^2}{4|C_2(s)|}$, where
>
> $C1(s) = ||{\nabla_{\theta_2}\mu_{\theta_2(s)}}||^2$
>
> $C_2(s) = \frac{1}{2} \left(\nabla_{\theta_2}\mu_{\theta_2}(s) \right)^\intercal H_{\mu_{\theta_2}}(s)\nabla_{\theta_2}\mu_{\theta_2}(s)\left(\nabla_{a_2}Q_2(s,a_1,a_2)\right)^\intercal$
>
>
> Based on Theorem 3, the projection estimation via first-order Taylor expansion and sufficiently small $\hat{\eta}_{\text{1st}}$ only scales the prediction length as both $\eta$ and $\eta'$ are non-negative numbers. The amount of this scaling depends on both values of $C_1(s)$ and $C_2(s)$.
>
> The general theoretical analyses on single-sate games reveal that scaling the prediction length directly influences the HOG methods' convergence speed [3],[4]. However, by directly changing the projected prediction length, i.e., $\hat{\eta}\_{\text{1st}}$, we can tune the resulting prediction length in the state space, i.e., $\eta'$, and consequently improve the convergence behavior. To empirically show this, we conducted an experiment to study the influence of $\hat{\eta}\_{\text{1st}}$ on the convergence behavior of LOLA-MADDPG in the Exit-room game (See Figure 7 in the revised manuscript). We show that by increasing $\hat{\eta}\_{\text{1st}}$ we can improve the convergence behavior of LOLA-MADDPG. However, high values of the projected prediction length can lead to instability of the algorithm which can be attributed to our findings in Theorem 3.
>
> To answer the Reviewer's questions, our projection estimation affects the prediction length, which influences the convergence behavior of the algorithms. However, there is a $\hat{\eta}\_{\text{1st}}$ for which the performance, i.e., convergence behavior, does not degrade, under the condition $\hat{\eta}\_{\text{1st}} < \frac{C_1(s)^2}{4|C_2(s)|}$.

---

> ### Author Response · Authors · 2022-11-14
> **Response to Reviewer eo4T [1/3]**
>
> We want to thank the Reviewer for the constructive feedback. With the responses provided below, we believe that we successfully address the major concerns of the reviewers regarding the lack of sufficient theoretical analyses in the manuscript.
>
> > 1. The review does not think that resorting to MADDPG can totally mitigate the computation difficulty of high-dimensional gradient. Even though  the high order gradient can be calculated, it takes long time when the gradient is of really high-dimensional.
>
> We want to thank the Reviewer for the comment. We agree with the Reviewer that resorting to MADDPG can not, in itself, mitigate the computation difficulty of high-dimensional gradients. To reduce the complexity of computing high-order gradients, however, we proposed to project the anticipated (high-order) gradients to a lower dimensional space, i.e., the action space. To give an example, the policy network in the Exit room game has around 2M parameters. Therefore, a first-order gradient of the state-action value with respect to the policy parameters is 2M-D, which is indeed high-dimensional. A single round of computing these high-dimensional first-order gradients is considered standard for an agent's policy update in stochastic and deterministic policy-based methods [1],[2]. However, multiple rounds of computing high-dimensional higher-order gradients, as should be done in the gradient anticipation, are anything but standard. This is while the action space in the Exit room game is 5-D. Therefore, by projecting the gradients from the parameter space to the action spaces, we significantly lower the dimensionality of the high-dimensional gradients, i.e., from 2M-D in the parameter space to 5-D in the action space.
>
> In order to clarify this better, we analyzed the time complexity of the gradient anticipation in Appendix B.2 of the revised manuscript. We show in Appendix B.2 that in the case of a fully connected policy network with $L$ layers and $N$ neurons in each layer, projection can reduce the time complexity by $O(L N^2)$ for a single round of gradient anticipation.

---

### Official Review · Reviewer_ynkz · 2022-10-24

**Confidence:** 3
**Correctness:** 3
**Technical Novelty And Significance:** 3
**Empirical Novelty And Significance:** 3
**Recommendation:** 6

**Clarity, Quality, Novelty And Reproducibility:**

Clear writing, good quality and novelty, need subsequent open source code and environment to verify its reproducibility


**Strength And Weaknesses:**

Strength.
* The authors have studied a very important direction
* The paper is clearly written and well understood
* A comprehensive review of existing papers so that the value of the methods can be pinpointed
* The experimental part looks adequate

Weaknesses.
* The authors claim that the existing methods are only applicable to low-dimensional settings, while their proposed methods can be applied to high-dimensional settings. But what puzzles me is that the experimental environment looks simple, while the number of real states in the environment is not very large, please note that I mean how many discrete states are in the environment.
* Although the experiments show the effectiveness of the authors' method, it looks like the variance is large and there is a relatively large overlap between the algorithms, does this mean that more iterations of the experiments are needed or think about other evaluation methods?


**Summary Of The Paper:**

The Higher-Order Gradient approach is of great value to the field of MAS, as it can be used in many important areas such as the theory of mind. The authors review the existing work and further propose Hierarchical Reasoning to facilitate the collaboration of team agents, noting that their approach is highly extensible and will help the community to explore the problem in the future.

**Summary Of The Review:**

Looks like a good paper, very interesting, I will adjust my score further after considering the author's rebuttal with other reviewers' comments.

---

> ### Author Response · Authors · 2022-11-14
> **Response to Reviewer ynkz [2/2]**
>
> > 2. Although the experiments show the effectiveness of the authors' method, it looks like the variance is large and there is a relatively large overlap between the algorithms, does this mean that more iterations of the experiments are needed or think about other evaluation methods?
>
> We thank the Reviewer for this comment. We want to highlight that we have repeated each experiment five times with random initialization. The variance in the results indicates the inconsistency of the methods' performances over all experimental runs rather than their convergence behaviors on each run. To clarify this, we provided the details of our experiment on the particle coordination game (Figure 2 in the manuscript) in the table below. The results per run are the means and standard deviations of the rewards (mean $\pm$ standard deviation) over the last 500 iterations of each run, and in the bottom row we provide the mean and standard deviation of the reward means over all five runs.
>
> |	Run		|      MADDPG      |  CPG-MADDPG | PR2-MADDPG| LOLA-MADDPG|LA-MADDPG|HR-MADDPG
> |:----------|:-------------:|:------:|:--:|:---:|:---:|:---:
> | 1 |  5.58 $\pm$ 0.01 | 4.74 $\pm$ 0.06 | 0.91 $\pm$ 0.05 | 5.17 $\pm$ 0.01 | 4.65 $\pm$ 0.03 | 4.96 $\pm$ 0.09
> | 2 |    0.91 $\pm$ 0.02  |   5.68 $\pm$ 0.02 | 0.90 $\pm$ 0.02 | -1.71 $\pm$ 0.01 | 0.67 $\pm$ 0.11 | 5.54 $\pm$ 0.03
> | 3 | 5.16 $\pm$ 0.03 |0.91 $\pm$ 0.01| 0.88 $\pm$ 0.01 | 2.20 $\pm$ 0.01 | 5.23 $\pm$ 0.02 | 5.55 $\pm$ 0.01
> | 4 | 0.92 $\pm$ 0.01|0.91 $\pm$ 0.02| 0.91 $\pm$ 0.01 | 4.68 $\pm$ 0.01 | 4.85 $\pm$ 0.07 | 5.65 $\pm$ 0.01
> | 5 |5.43 $\pm$ 0.02|5.55 $\pm$ 0.09| 5.55 $\pm$ 0.07 | 0.79 $\pm$ 0.09 |-3.48 $\pm$ 0.09 | 5.57 $\pm$ 0.01
> | AVG of reward means | 3.60 $\pm$ 2.20|3.56 $\pm$ 2.19| 1.77 $\pm$ 1.73 | 2.45 $\pm$ 2.56 | 2.39 $\pm$ 3.67 | **5.45 $\pm$ 0.25**
>
> As shown, all methods have low standard deviations on each run, indicating that they are fully converged. However, for all methods except our proposed HR-MADDPG, the standard deviations over all runs (as shown in Figure 2) are high, indicating inconsistent performances. The results reveal that not only does HR-MADDPG have consistent performance, but also it significantly outperforms other methods.
>
> We would like to thank the Reviewer again for taking the time to review our manuscript. We kindly ask the Reviewer to let us know if the responses are clear, or if the Reviewer desires any further clarifications.
>
>     [1] Chongjie Zhang and Victor Lesser. Multi-agent learning with policy prediction. In  Proceedings of the AAAI Conference on Artificial Intelligence, volume 24, 2010.
>     [2] Jakob Foerster, Richard Y Chen, Maruan Al-Shedivat, Shimon Whiteson, Pieter Abbeel, and Igor Mordatch. Learning with Opponent-Learning Awareness. In Proceedings of the 17th International Conference on Autonomous Agents and MultiAgent Systems, pp. 122–130, 2018a.

---

> > ### Comment · Reviewer_ynkz · 2022-11-18
> > **Reply To The Authors**
> >
> > Dear Authors.
> >
> > Thank you for your detailed replies, the current ones solved my confusion. I will consider the comments of other reviewers and then further consider my decision.

---

> ### Author Response · Authors · 2022-11-14
> **Response to Reviewer ynkz [1/2]**
>
> We want to thank the Reviewer for the constructive feedback. We are encouraged by the comments that the studied direction is very important, and that the paper is clearly written. Below, we individually address all of the Reviewer's concerns and questions.
>
>
> > 1. The authors claim that the existing methods are only applicable to low-dimensional settings, while their proposed methods can be applied to high-dimensional settings. But what puzzles me is that the experimental environment looks simple, while the number of real states in the environment is not very large, please note that I mean how many discrete states are in the environment.
>
> We thank the Reviewer for the comment. We want to highlight that current HOG methods are only applied in games with discrete-state spaces, such as iterated rotational game and iterated prisoner's dilemma. Consequently, the parameter space proposed in [1] and [2] is low-dimensional. More specifically:
>
> |			|      State space (type)      |  # Discrete states | Parameter space
> |----------|:-------------:|:------:|:--:
> | Iterated rotational game |  1-D (discrete) | 1 | 1
> | Iterated prisoner's dilemma |    1-D (discrete)  |   5 | 5
>
> On the other hand, we proposed the HOG-MADDPG framework to handle continuous-state spaces where the states are real-valued vectors. Therefore, the number of discrete states is infinite, resulting in high-dimensional parameter spaces. More specifically:
>
> |			|      State space (type)     |  # Discrete states | Parameter space
> |----------|:-------------:|:------:|:--:
> | Exit-room game |  225-D (continuous) | Infinite (12M active) | 2M-D
> | Particle coordination game |    14-D (continuous)  |   Infinite | 11K-D
> | Cooperative navigation | 18-D (continuous)|Infinite| 11K-D
> | Physical deception | 10_D (continuous)|Infinite| 11K-D
> | Predator-prey |14-D (continuous)|Infinite| 10K-D
> | Half-cheetah | 11-D (continuous)|Infinite| 70K-D
> | Walker | 11-D (continuous)|Infinite| 70K-D
> | Reacher | 8-D (continuous)|Infinite| 70K-D
>
> We added this information to Table 6 of the revised manuscript.

---

### Official Review · Reviewer_VXeR · 2022-10-26

**Confidence:** 3
**Correctness:** 3
**Technical Novelty And Significance:** 3
**Empirical Novelty And Significance:** Not applicable
**Recommendation:** 5

**Clarity, Quality, Novelty And Reproducibility:**

The writing and presentation of the paper could be improved. The authors should also further highlight the key novelty of the proposed algorithms and explain more about their advantages compared to existing methods.

**Strength And Weaknesses:**

Strength
- The problem in consideration is interesting.
- The proposed algorithm performs well in the experiments.

Weakness
- The paper could make it clear what the exact setting is. This will help clarify some potential confusion. For example, the issue raised in the intro does not seem to exist in the CTDE framework (not a game setting).
- What are the forms of (1) and (2) when there are multiple agents? So are (3)-(7). Right now everything is done with n=2.
- Following the above comment, it appears that most of the derivation and most experiments are done for a 2-user case (or two coordinating agents with others). While the reviewer understands the tractability of this case, it seems a bit limited that even in the experiments only two agents are considered.
- Where is the projection step in the algorithms? Also, what is the complexity for doing so?
- The presentation is a bit confusing. In Fig. 2,  LA-MADDPG and LOLA-MADDPG are both marked as “ours”. But in Table 3, only HR-MADDPG has this label. From the text, it seems that the paper applies the ideas to LA-MADDPG and LOLA-MADDPG. Please clarify.
- It would be useful to highlight the key difference between the proposed algorithms and MADDPG. Right now the novelty of the algorithm is not clear.
- Minor: “between the agents” should be “among the agents”

**Summary Of The Paper:**

This paper studies the use of higher order gradient methods for multi-agent RL with high dimensional state space. It shows that existing methods can lead to miscoordination among agents. A hierarchical reasoning algorithm is proposed. Experimental results are presented to show the applicability of the method.



**Summary Of The Review:**

Overall, the paper considers an interesting problem and the experimental results show that the algorithms perform well. However, the presentation of the paper could be improved, also, the contributions need to be better explained.

---

> ### Author Response · Authors · 2022-11-14
> **Response to Reviewer VXeR [1/4]**
>
> We want to thank the Reviewer for the constructive feedback. We are encouraged by the comments that the addressed problems are interesting, and that our proposed algorithm performs well. Below, we individually address all of the Reviewer's concerns and questions.
>
> > 1. Overall, the paper considers an interesting problem and the experimental results show that the algorithms perform well. However, the presentation of the paper could be improved, also, the contributions need to be better explained.
>
> We want to thank the Reviewer for the constructive comments. Below,  we address all of the Reviewer's concerns individually. To improve our paper's presentation and clarity, we added additional explanations and a contributions list to the revised manuscript (at the end of Section 1). These are the main contributions of our paper:
>
>  - We propose HOG-MADDPG, a framework to make existing HOG methodologies, e.g., LA and LOLA, applicable to games with higher-dimensional state spaces by solving the limitations in computation and preservation of higher-order gradient information. With our framework, we develop two novel HOG methods, LA-MADDPG and LOLA-MADDPG, which apply the principles of  LA and LOLA, respectively.
>  - We demonstrate theoretically, in a two-agent two-action coordination game, and empirically, in a two-agent three-action coordination game, that the existing HOG methodologies can suffer from miscoordination among common-interested agents. To solve this, we propose the HR methodology and show, theoretically and empirically, that it overcomes miscoordination in the coordination games.
>  - We apply the HR principle to our HOG-MADDPG framework and develop HR-MADDPG, a HOG method for common-interested agents. We show that HR-MADDPG outperforms the existing state-of-the-art methods on standard multi-agent games.
>
> > 2. The paper could make it clear what the exact setting is. This will help clarify some potential confusion. For example, the issue raised in the intro does not seem to exist in the CTDE framework (not a game setting).
>
> We agree with the Reviewer that more clarification is needed regarding the exact setting of our framework. Our paper follows the Centralized Training and Decentralized Execution (CTDE) setting, and we clarified this in our revised manuscript (Section 4.2).
>
> CTDE is a standard setting in multi-agent planning where agents can share information during training. However, CTDE does not imply any constraints on information sharing and, more importantly, it does not impose any rules on the way the gradient information is computed or preserved, which are the issues raised in our paper. To clarify this, we emphasize the key differences between our proposed LOLA-MADDPG and the latest proposed HOG method, LOLA-DiCE [1]. We show that although both methods follow the CTDE settings, LOLA-DiCE requires higher information-sharing levels and does not solve the raised issues of gradient computation and preservation.
>
>  1. **Gradient computation (first issue):** LOLA-DiCE is implemented in the stochastic policy gradient framework and optimizes non-differentiable objectives. Given the CTDE setting, the agents first access the opponents' information and reason about the opponents' learning steps in inner (reasoning) learning loops. Then, the agents update their parameters in outer learning loops. However, as the agents optimize non-differentiable objectives, they require sampling stages for both inner and outer learning loops. This is very inefficient for high-order reasoning and games with higher-dimensional state spaces. In LOLA-MADDPG, on the other hand, agents optimize centralized differentiable objective functions and do not need sampling for each reasoning (inner) loop.
>  2. **Gradient perservation (second issue):**  In LOLA-DiCE, each agent should keep track of all agents' higher-order gradient values and computation graphs in the shared policy parameter spaces, which is inefficient for games requiring high-dimensional parameter spaces. In LOLA-MADDPG, on the other hand, the higher-order gradient information is preserved in the action space, which has a significantly lower dimension than the parameter space.
>  3. **Information-sharing level:** Following 3, the agents in LOLA-DiCE have access to opponents' policy parameters. In LOLA-MADDPG, on the other hand, the agents cannot access opponents' policy parameters and can only see the opponents' actions. This constraint on information sharing is of primary importance in games with higher-dimensional state spaces, e.g., images as the states. These games require deeper policy networks, and, consequently, the parameter space dimension can be significantly higher than the action space dimension.
>
> Therefore, because these issues occur in CTDE-based methods like LOLA-DiCE, we conclude that they can occur in the CTDE setting, and that our framework is necessary to solve it.

---

> ### Author Response · Authors · 2022-11-14
> **Response to Reviewer VXeR [2/4]**
>
> > 3. What are the forms of (1) and (2) when there are multiple agents? So are (3)-(7). Right now everything is done with n=2.
>
> We agree with the Reviewer that equations in their current forms could imply that the methods are only applicable to two agents. We first want to highlight that the two-agent assumption is only for the sake of presentation simplicity, and such an assumption has also been considered in previous works [2],[4].
>
> Regarding equations (1) and (2) in the main manuscript, we followed the two-agent assumption of our primary references for LOLA [2] and LA [4] to provide sufficient background about HOG methods. However, as discussed in [2],[3], both methods are generalizable to games with more than two agents. For instance, the LA update for the case of three agents is:
>
> $\nabla_{\theta_1}V_1^{\text{LA}}(s,\theta_1,\theta_2+\Delta\theta_2,\theta_3+\Delta\theta_3) \approx \nabla_{\theta_1}V_1 + (\nabla_{\theta_2\theta_1}V_1)^\intercal \Delta\theta_2+ (\nabla_{\theta_3\theta_1}V_1)^\intercal \Delta\theta_3$
>
> In Equations (3-7), we describe our proposed LA-MADDPG and LOLA-MADDPG for the case of two agents. For the general case of $n$ agents for LA-MADDPG and LOLA-MADDPG, we refer to algorithms 1 and 2 in Appendix A of the revised manuscript, respectively. For instance, the LOLA-MADDPG update for the case of three agents is:
>
> $\nabla_{\theta_1}J_1^{\text{LOLA}} \approx \mathbb{E}\_{\rho^\beta(s, a)} \nabla_{\theta_1}\mu_{\theta_1}(s)\nabla_{a_1}Q_1(s,a_1,a_2+\Delta a_2,a_3+\Delta a_3)|\_{a_1=\mu_{\theta_1}(s)}$
>
> where $\Delta a_2 = \hat{\eta}\nabla_{a_2}Q(s,a_1,a_2,a_3)$ and $\Delta a_3 = \hat{\eta}\nabla_{a_3}Q(s,a_1,a_2,a_3)$.
>
> Additionally, we added the LOLA-MADDPG and LA-MADDPG updates for the general case of $n$ agents in Appendices A.1 and A.2 of the revised manuscript, respectively.
>
> > 4. Following the above comment, it appears that most of the derivation and most experiments are done for a 2-user case (or two coordinating agents with others). While the Reviewer understands the tractability of this case, it seems a bit limited that even in the experiments only two agents are considered.
>
> In our paper, as mentioned by the Reviewer, we limited the experiments to games with a maximum of three agents ($n=3$): three cooperative agents in the Cooperative Navigation game, two cooperative agents and one competitive agent in the Physical Deception and Predator-Prey games, and two competitive agents in the Mujoco games. We want to highlight that our proposed HOG-MADDPG framework is developed on top of the MADDPG method, and this limitation on the number of agents is inherent to the MADDPG method rather than the proposed HOG methodology. More specifically, the limitation comes from the core concept of centralized state-action value functions in the MADDPG method. As argued by Peng et al. [5], a centralized value function cannot scale well when the number of agents increases ($n>3$). Consequently, all methods in the literature developed in this platform, such as [6] and [7], have limited the experiments to three agents. To overcome this, Peng et al. [5] applied the concept of the centralized, but *factorized* value functions on MADDPG. They proposed a method, referred to as FACMAC, that is more scalable for many-agent settings. We believe that employing FACMAC instead of MADDPG, as the basis of our framework, is a potential direction for future work. We clarified this direction in the Discussion of the revised manuscript (Section 6).
>
> > 5. Where is the projection step in the algorithms? Also, what is the complexity for doing so?
>
>
> We thank the Reviewer for pointing out this important comment. In our paper, we analytically derive the projection estimation step. Consequently, we employ the analytical form of the projection estimation step in our algorithms. We have now indicated this projection step in all Algorithms of the revised manuscript. Additional theoretical analyses for this projection step are provided in Appendix B.1 of the revised manuscript.
>
> Furthermore, we analyzed the time complexity of the proposed projection estimation, and we added our findings in Appendix B.2 of the revised manuscript. We show in Appendix B.2 that by projecting the anticipated gradients to the action space, the time complexity of gradient anticipation is reduced by $O(L N^2)$, where $L$ and $N$ are the numbers of layers and neurons per layer in a fully connected policy network, respectively.

---

> ### Author Response · Authors · 2022-11-14
> **Response to Reviewer VXeR [3/4]**
>
> > 6. The presentation is a bit confusing. In Fig. 2, LA-MADDPG and LOLA-MADDPG are both marked as “ours”. But in Table 3, only HR-MADDPG has this label. From the text, it seems that the paper applies the ideas to LA-MADDPG and LOLA-MADDPG. Please clarify.
>
> We understand that the presentation of the figures and tables cause some confusion to the Reviewer regarding the contributions of our paper. In Table 3, we specified "ours" only for HR-MADDPG because the purpose of this experiment is to show the benefit of HR-MADDPG with respect to other MADDPG-based methods, including LOLA-MADDPG and LA-MADDPG, which we proposed earlier in our work for other types of games. To clarify: in our paper, we developed three novel HOG methods: LA-MADDPG, LOLA-MADDPG, and HR-MADDPG. All three methods are now marked as "ours" in all Figures and Tables of the revised manuscript.
>
> As mentioned in the contributions list of the revised manuscript, we first propose the HOG-MADDPG framework to make existing HOG methodologies applicable to games with higher-dimensional state spaces. This is done by solving the raised issues regarding the computation and preservation of the higher-order gradient information. Afterwards, we employed our framework to develop two novel HOG methods, LA-MADDPG and LOLA-MADDPG, which use the LA and LOLA principles, respectively. Furthermore, we proposed a novel HOG methodology, referred to as HR, to increase coordination among common-interested agents. Finally, we applied the HR principle to our proposed HOG-MADDPG framework and developed our third HOG method, i.e., HR-MADDPG. We clarified the contributions in the revised manuscript.
>
> > 7. It would be useful to highlight the key difference between the proposed algorithms and MADDPG. Right now the novelty of the algorithm is not clear.
>
> We thank the Reviewer for the comment. The critical difference is that our proposed methods reason about the other agents' anticipated learning using Higher-Order Gradient (HOG) information, and that standard MADDPG methods do not reason about other agents at all. We clarified this in Section 4.2 of the revised manuscript. Here we provide more details.
>
> In MADDPG [6] (as well as CPG-MADDPG [5]), agents do not reason about the anticipated learning of other agents. In other words, the agents are naive learners and use standard gradient descent algorithms to optimize the policy parameters. On the other hand, the agents in our proposed methods, i.e., LA-MADDPG, LOLA-MADDPG, and HR-MADDPG, use various HOG principles (i.e., LA, LOLA, and HR) to anticipate the learning steps of other agents and update the policy parameters. More specifically:
>
>  - Our LA-MADDPG method is developed on top of the MADDPG method and employs the LA principle to reason about other agents' anticipated learning.
>  - Our LOLA-MADDPG method is developed on top of the CPG-MADDPG method and employs the LOLA principle to reason about other agents' anticipated learning.
>  - Our HR-MADDPG method is developed on top of the CPG-MADDPG method and employs our proposed HR principle to reason about other agents' anticipated learning.

---

> ### Author Response · Authors · 2022-11-14
> **Response to Reviewer VXeR [4/4]**
>
> > 8. Minor: “between the agents” should be “among the agents”
>
> We thank the Reviewer for this suggestion. We corrected the term in the revised manuscript.
>
> We would like to thank the Reviewer again for taking the time to review our manuscript. We kindly ask the Reviewer to let us know if the responses are clear, or if the Reviewer desires any further clarifications.
>
>
> 	[1] Jakob Foerster, Gregory Farquhar, Maruan Al-Shedivat, Tim Rocktaschel, Eric Xing, and Shimon Whiteson. Dice: The infinitely differentiable monte carlo estimator. In International Conference on Machine Learning, pp. 1529–1538. PMLR, 2018c.
> 	[2] Jakob Foerster, Richard Y Chen, Maruan Al-Shedivat, Shimon Whiteson, Pieter Abbeel, and Igor Mordatch. Learning with Opponent-Learning Awareness. In Proceedings of the 17th International Conference on Autonomous Agents and MultiAgent Systems, pp. 122–130, 2018a.
> 	[3] Alistair Letcher, Jakob Foerster, David Balduzzi, Tim Rocktaschel, and Shimon Whiteson. Stable ¨ Opponent Shaping in Differentiable Games. In International Conference on Learning Representations, 2019.
> 	[4] Chongjie Zhang and Victor Lesser. Multi-agent learning with policy prediction. In Proceedings of the AAAI Conference on Artificial Intelligence, volume 24, 2010.
> 	[5] Bei Peng, Tabish Rashid, Christian Schroeder de Witt, Pierre-Alexandre Kamienny, Philip Torr, Wendelin Boehmer, and Shimon Whiteson. FACMAC: Factored Multi-Agent Centralised Policy Gradients. NeurIPS, 2021.
> 	[6] Ryan Lowe, Yi Wu, Aviv Tamar, Jean Harb, Pieter Abbeel, and Igor Mordatch. Multi-agent actor-critic for mixed cooperative-competitive environments. In  NIPS, 2017.
> 	[7] Ying Wen, Yaodong Yang, Rui Luo, Jun Wang, and Wei Pan. Probabilistic recursive reasoning for multi-agent reinforcement learning.  In  7th International Conference on Learning Representations, ICLR 2019, 2019.

---

### Author Response · Authors · 2022-11-14
**General Response to all Reviewers**

We thank all the Reviewers for their constructive comments, which have helped us to improve our paper. With detailed individual responses to each Reviewer, we believe that we adequately addressed all their concerns and questions. Overall, we applied the following changes in our revised manuscript:



1.  We clarified the main contributions of our paper in Section 1.
2.  We clarified the primary setting of our HOG-MADDPG framework and stated the critical difference with MADDPG in Section 4.2.
3.  We clarified our future direction for many-agent games in Section 6.
4. We added the LOLA-MADDPG and LA-MADDPG updates for the general case of $n$ agents in Appendices A.1 and A.2, respectively.
7. We clarified the projection estimation step in all algorithms in Appendices A.1, A.2, and A.3.
8. We provided theoretical analyses on the influence of projection estimation in Appendix B.1.
9. We analyzed the time complexity of the projection estimation in Appendix B.2.

---

### Decision · Program_Chairs · 2023-01-20

**Decision:**

Reject

**Justification For Why Not Higher Score:**

The paper's presentation is not clear and it is very difficult to follow the authors' ideas. A complete rewrite would be needed before reconsidering this paper for inclusion in the technical program of a top-tier ML conference.

**Justification For Why Not Lower Score:**

N/A

**Metareview: Summary, Strengths And Weaknesses:**

This paper concerns the use of higher-order gradient (HOG) methods in the context of multi-agent reinforcement learning (MARL). The bottleneck that the authors seek to tackle is the intractability of higher-order derivative computations in problems with high-dimensional state spaces (where "higher-order" refers primarily to the cross-terms that appear in gradient updates when attempting to take into account the actions of other players). To that end, they take as a starting point the multi-agent deep deterministic policy gradient (MADDPG) template of Lowe et al. (2017), and they exploit the available information to anticipate for changes in the agent's policies and actions. This policy is subsequently evaluated experimentally in an iterated prisoner's dilemma and team games (where the authors also study the possibility of miscoordination as a function of the prediction length and the miscoordination parameter of the game).

The reviewers raised several concerns which were only partially addressed by the authors. In particular, even though the authors revised their original submission, the presentation remained unclear (as an example, the proposed algorithms are only described at a very high level in the main body of the paper, so it becomes very difficult to follow the authors' ideas), and the paper's main contributions were found somewhat incremental (one criticism being the lack of a theoretical complexity analysis that could clarify the gains of the proposed hierarchical approach).

Overall, the conclusion of the discussion phase was that the paper does not meet the acceptance criteria for ICLR, so a decision was reached to make a "reject" recommendation to the program committee.

**Summary Of Ac-Reviewer Meeting:**

Even though the scores of the paper were borderline, a conclusion was quickly reached that the paper does not meet the acceptance criteria for ICLR.